# Design principles of transcription factors with intrinsically disordered regions

**Wencheng Ji**[1,2]*, **Ori Hachmo**[1], **Naama Barkai**[3], **Ariel Amir**[1,2]*

[1]Department of Physics of Complex Systems, Weizmann Institute of Science, Rehovot, Israel; [2]John A Paulson School of Engineering and Applied Sciences, Harvard University, Cambridge, United States; [3]Department of Molecular Genetics, Weizmann Institute of Science, Rehovot, Israel

## eLife Assessment

This paper presents an **important** theoretical exploration of how a flexible protein domain with multiple DNA binding sites may simultaneously provide stability to the DNA-bound state and enables exploration of the DNA strand. The authors propose a mechanism ("octopusing") for protein doing a random walk while bound to DNA which simultaneously enables exploration of the DNA strand and enhances the stability of the bound state. This study presents **compelling** evidence that their findings has implications for the way intrinsically disordered regions (IDR) of transcription factors proteins (TF) can enhance their ability to efficiently find their binding site on the DNA from which they exert control over the transcription of their target gene. The paper concludes with a comparison of model predictions with experimental data which gives further support to the proposed model.

**\*For correspondence:**
wencheng.ji@weizmann.ac.il
(WJ);
ariel.amir@weizmann.ac.il (AA)

**Abstract** Transcription factors (TFs) are proteins crucial for regulating gene expression. Effective regulation requires the TFs to rapidly bind to their correct target, enabling the cell to respond efficiently to stimuli such as nutrient availability or the presence of toxins. However, the search process is hindered by slow diffusive movement and the presence of 'false' targets – DNA segments that are similar to the true target. In eukaryotic cells, most TFs contain an intrinsically disordered region (IDR), which is commonly assumed to behave as a long, flexible polymeric tail composed of hundreds of amino acids. Recent experimental findings indicate that the IDR of certain TFs plays a pivotal role in the search process. However, the principles underlying the IDR's role remain unclear. Here, we reveal key design principles of the IDR related to TF binding affinity and search time. Our results demonstrate that the IDR significantly enhances both of these aspects. Furthermore, our model shows good agreement with experimental results, and we propose further experiments to validate the model's predictions.

## Introduction

Transcription factors (TFs) are fundamental proteins that regulate gene expression: they bind to specific DNA sequences to control gene transcription *Latchman, 1997*; they respond adaptively to various cellular signals, thereby modulating gene expression *Mitchell and Tjian, 1989*; *Ptashne and Gann, 1997*; they guide processes such as cell differentiation and development, and their malfunctions are often linked to disease onset (*Lambert et al., 2018*; *Stadhouders et al., 2019*). The principles of gene regulation were first formulated in the 1960s through pioneering studies of the *E. coli* Lac operon (*Jacob and Monod, 1961*), establishing that the regulation mechanism is via the binding

of a TF to the regulatory DNA regions associated with the genes, the TF's target. Effective regulation requires TFs to have sufficient binding affinity and a binding rate comparable to cellular processes. Despite these decades of research, the factors determining the binding strength and search efficiency of eukaryotic TFs remain incompletely understood (*Ferrie et al., 2022*; *Jonas et al., 2025*). In eukaryotic cells, besides having a DNA-binding domain (DBD), approximately 80% of TFs also have one or more intrinsically disordered regions (IDRs), which lack a stable 3D structure and can be assumed as flexible, unstructured polymers, hundreds of amino acids (AAs) in length (*Liu et al., 2006*; *Fuxreiter et al., 2011*; *Guo et al., 2011*; *Wright and Dyson, 2015*). Eukaryotic DBDs often exhibit weaker affinity to their recognition sequences compared to their bacterial counterparts, indicating that IDRs are necessary for achieving stable binding (*Garcia et al., 2021*). However, IDRs are challenging to study because their functions can be maintained despite rapid sequence divergence, complicating traditional sequence-based analyses (*Brown et al., 2010*; *Zarin et al., 2019*). Molecular dynamics simulations suggest that IDRs play a significant role in promoting target recognition (*Vuzman and Levy, 2010*), by increasing the area of effective interaction between the TF and the DNA. Upon binding, the search problem becomes effectively a one-dimensional diffusion process (*Vuzman and Levy, 2012*). Recent in vivo experiments done in yeast also suggest that IDRs could take a key part in guiding TFs to the DNA regions containing their targets (*Brodsky et al., 2020*; *Kumar et al., 2023*). Truncation studies of IDRs in various TFs have demonstrated that removing these regions often results in reduced binding affinity, suggesting they contribute significantly to stabilizing TF-DNA interactions (*Brodsky et al., 2020*). Different mechanisms have been proposed in the literature to explain IDR contributions: some suggest non-specific electrostatic interactions (*Iwahara et al., 2006*), others propose phase separation that concentrates TFs near their targets (*Boija et al., 2018*), while a third view suggests specific IDR-DNA interactions analogous to those of structured domains (*Chappleboim et al., 2024*).

The problem of TF search has been widely studied, theoretically, albeit with the majority of works focusing on parameter regimes relevant for bacteria. One important mechanism is that of facilitated diffusion, in which the TF performs 1D diffusion along the DNA and at any given moment can fall off and perform 3D diffusion until reattaching to the DNA (a 3D excursion). The 1D diffusion along the DNA and the 3D excursions alternate until the TF finds its target. This mechanism is well characterized

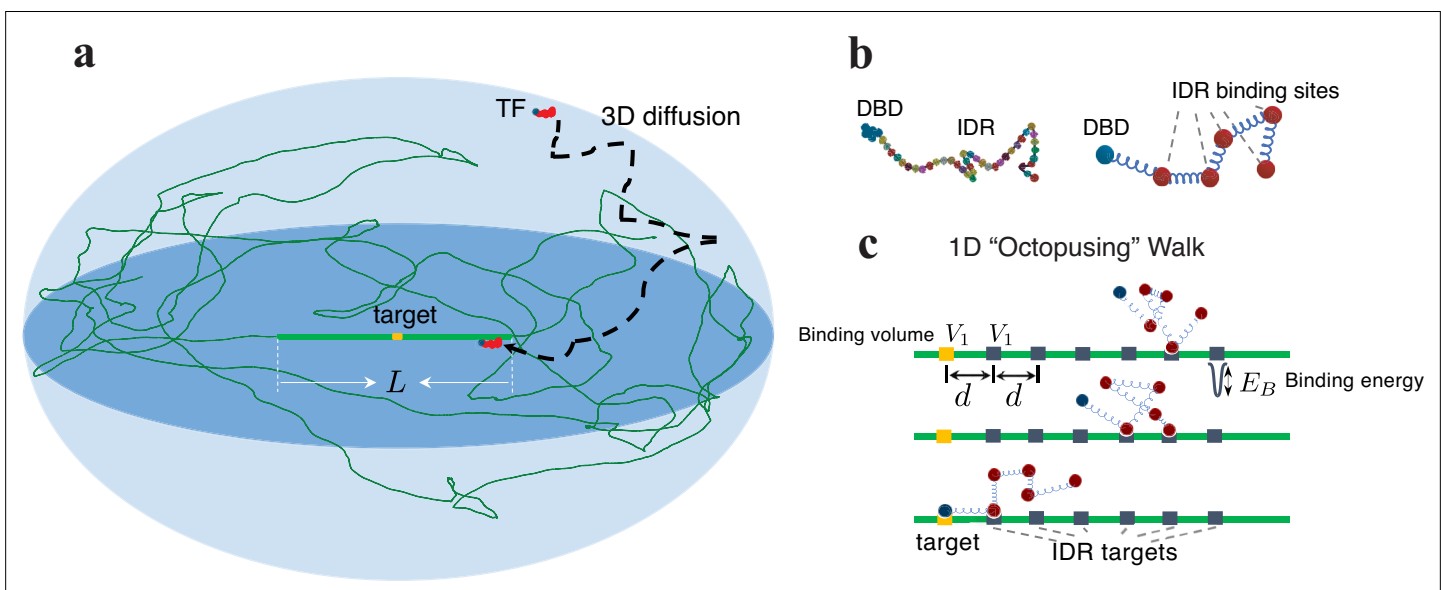

**Figure 1.** Model illustration of a TF with the IDR and its search process. (**a**) Illustration of a TF locating its target site (highlighted in orange) on the DNA strand (depicted as a green curve). Surrounding the target is a region of length $L$, where the TF's IDR interacts with the DNA. The TF performs 3D diffusion until it encounters and binds to this 'antenna' region. (**b**) Illustration of a TF that includes an IDR composed of AAs. On the right: a model of the TF, a polymer chain, comprising a single binding site on the DNA Binding Domain (DBD), and multiple binding sites on the IDR (also known as short linear motifs *Jonas et al., 2025*). (**c**) Once bound to the antenna, the TF performs effective 1D diffusion until it reaches its target. The 1D diffusion is via the binding and unbinding of sites along the IDR, a process we coin 'octopusing'. $V_1$ is the targets' volume, $d$ the separation, and $E_B$ the binding energy per binding site, where each site corresponds to a short linear motif.

theoretically (*Berg et al., 1981*; *Berg and von Hippel, 1987*; *Slutsky and Mirny, 2004*; *Mirny et al., 2009*; *Li et al., 2009*; *Bénichou et al., 2011*; *Sheinman et al., 2012*) and supported by experiments in *E. coli* (*Hammar et al., 2012*). However, due to the significant structural differences, including larger genome size and chromatin structure, the facilitated diffusion mechanism cannot explain the fast search times observed in eukaryotes, and a qualitatively different mechanism is required (*Dror et al., 2016*; *Jana et al., 2021*; *Ferrie et al., 2022*; *Kumar et al., 2023*; *Wagh et al., 2023*).

Here, we attempt to address the fundamental question of how IDRs can enhance the binding affinity of the TFs and speed up their search process and to discover the associated design principles in eukaryotes. The problem of binding affinity (binding probability) exhibits the classical trade-off between energy and entropy: on one hand, the binding of IDR to the DNA is energetically favorable. On the other hand, this binding leads to constraints on the positions of the IDR (pinned to the DNA at the points of attachment), which leads to a reduction in the entropy associated with the possible polymer conformations – hence an *increase* in the free energy of the system. Therefore, the effects of the disordered tail on affinity are non-trivial, leading to physical problems reminiscent of those associated with the well-studied problem of DNA unzipping (*Danilowicz et al., 2004*). We find that the binding probability increases dramatically for TFs having IDRs for a broad parameter regime. The optimal search process consists of one round of 3D-1D search, where at first the TF performs 3D diffusion and then binds the antenna and performs 1D diffusion until finding its target, rarely unbinding for another 3D excursion. This process is presented in *Figure 1a and c*. We obtain an expression for the search time that shows good agreement with our numerical results.

## Results

### Binding probability of the TF from the equilibrium approach

The experiment described in *Brodsky et al., 2020* studied the binding affinity of wild-type and mutated TFs to their corresponding targets. They found that shortening the IDR length results in weaker affinity. To qualitatively model the binding affinity of a TF to its target, we utilize a well-known physical model of polymers, the Rouse Model (*Prince, 1953*; *Doi and Edwards, 1988*; used in the subsequent dynamical section as well). See Materials and methods. We note that our conclusions do not hinge on this choice, and our results hold also when using other polymer models (see Appendix Appendix 1). Broad applicability of probability density $f(r, l)$. Since the IDR is thought to recognize a relatively short flanking DNA sequence of the DBD target (*Brodsky et al., 2020*), we model a DNA segment as a straight-line 'antenna' region (*Shimamoto, 1999*; *Hu and Shklovskii, 2007*; *Mirny et al., 2009*; *Castellanos et al., 2020*), where the IDR targets reside. We assume that the interaction between TF and DNA outside the antenna region is negligible. Note that the interaction between IDR and DNA is a specific interaction, with genomic localization determined by multivalent interactions mediated by hydrophobic residues (*Jonas et al., 2023*; *Mindel et al., 2024*).

To gain an intuition into the binding probability of TF, we first study a TF with a single IDR site. From here on, energies will be measured in units of $k_B T$, to simplify our notation. We assume that the IDR has only one corresponding target. There are four possible states: neither the DBD nor the IDR is bound, the IDR is bound, the DBD is bound, and both are bound, which correspond to $P_{\text{free}}$, $P_{\text{IDR}}$, $P_{\text{DBD}}$, and $P_{\text{both}}$, respectively. The probability that the TF is bound at the DBD target is the sum of $P_{\text{DBD}}$ and $P_{\text{both}}$. In thermal equilibrium, each of the 4 terms is given by the Boltzmann distribution, which can be solved analytically by integrating out the additional degrees of freedom associated with the polymer (Appendix 2. Derivation of the binding probability). To better understand the associated design principles, it will be helpful to compare this probability with that of a 'simple' TF (i.e. without the IDR), which we denote by $P_{\text{simple}}$. We find that:

$$\frac{P_{\text{TF}}}{P_{\text{simple}}} \approx (1 + \mathcal{P}), \tag{1}$$

where $\mathcal{P}$ is an enhancement factor, governing the degree to which the IDR improved the binding affinity. Importantly, we find that:

$$\mathcal{P}(E_B, d, l_0) = e^{E_B} V_1 f(d, l_0), \tag{2}$$

where $f(d, l_0) = \left(\frac{3}{2\pi l_0^2}\right)^{3/2} \exp\left(-\frac{3d^2}{2l_0^2}\right)$, is the probability density, with units of inverse volume, $V_1$ is the targets' volume, on the order of $1 - 10\text{bp}^3$, as shown in *Figure 1c*, $-E_B$ is the binding energy for per IDR target ($E_B$ is always positive), $d$ is the distance between the DBD target and the IDR target, and $l_0$ is the distance between the DBD and the IDR binding site on the TF ($l_0$ and $d$ are also the distances for neighboring IDR targets and neighboring IDR sites, respectively, in the case of multiple IDR targets and sites).

This shows that the presence of the IDR always increases the binding affinity since $\mathcal{P}$ is always positive. However, the magnitude of enhancement can be negligible: For instance, even if $E_B$ is large, the term $f(d, l_0)$, associated with the entropic penalty (because the end-to-end distance is fixed, reducing the degrees of freedom compared to a free polymer), could lead to small values of $\mathcal{P}$.

The exponential decay of $\mathcal{P}$ with $d^2$ (through the term $f(d, l_0)$), indicates that placing the IDR targets close to the DBD target is preferable. However, due to physical limitations, $d$ cannot be arbitrarily small and in reality will have a lower bound. Once $d$ is fixed, $\mathcal{P}$ is maximal at $l_0 = d$, as can be gleaned from *Equation 2*. This condition minimizes the free energy penalty associated with the entropic contribution of the term $f(d, l_0)$.

As we shall see, this design principle is also intact when multiple IDR sites and targets are involved, which is the scenario we discuss next.

A *priori*, it is unclear whether having more binding sites ($n_b > 1$) on the IDR or more targets ($n_t > 1$) on the DNA will be beneficial or not. We therefore sought to investigate how the binding affinity depends on these parameters, as well as the binding energy and positions of the targets and binding sites. This generalization requires taking into account all the possible number of bindings, $n$ (ranging from 0 to $\min(n_b, n_t)$), and all possibilities for binding orientations for each $n$, $\binom{n_b}{n}\binom{n_t}{n}n!$ distinct configurations in total. For each of them, we may analytically integrate out the polymer degrees of freedom and calculate explicitly the contribution to the partition function (Appendix 2. Derivation of the binding probability). Summing these terms together, numerically, allows us to explore how the different parameters affect the binding specificity, which we describe next. To characterize the binding affinity, we resort again to the ratio $Q \equiv P_{\text{TF}}/P_{\text{simple}}$, measuring the effectiveness of the binding in comparison to the case with no IDRs. The length of TFs in our considerations, which varies with $n_b$, is comparable to that in experiments (*Brodsky et al., 2020*).

We find the design principle: $l_0 = d$, identical to what we showed for the case of a single IDR binding site. Specifically, $Q$ is maximal at $l_0 \sim d$ for all combinations of $n_b$ and $n_t$. *Figure 2a* displays this principle for several combinations of $n_b$ and $n_t$. All combinations are provided in *Appendix 3— figure 1*. This is plausible since for the IDR to be bound to the DNA at multiple places without paying a large entropic penalty, the distance between target sites should be compatible with the distance between binding sites.

We therefore set $l_0 = d$ and search for other design principles. *Figure 2b* shows a heatmap of $Q$ varying with $n_b$ and $n_t$. At large $n_t$, as $n_b$ increases, the value of $P_{\text{TF}}$ exhibits an initial substantial increase, followed by a gradual rise until it reaches saturation (shown explicitly in *Appendix 4—figure 1*).

A similar behavior is also observed when varying the value of $n_t$. (When $n_t$ exceeds $n_b$, the contribution from the states where the IDR is bound but the DBD is unbound increases with increasing $n_t$, which slightly lowers $P_{\text{TF}}$; see *Appendix 4—figure 1*.)

In fact, the heatmap shown in *Figure 2b* shows that the value of $P_{\text{TF}}$ is governed, approximately, by the *minimum* of $n_b$ and $n_t$ (this is reflected in the nearly symmetrical 'L-shapes' of the contours). This places strong constraints on the optimization and suggests that there will be little advantage in increasing the value of either $n_b$ or $n_t$ beyond the point $n_b = n_t$, revealing another design principle (under the assumption that it is preferable to have smaller values of $n_b$ and $n_t$ for a given level of affinity, in order to minimize the cellular costs involved).

We verify this principle holds at additional values of $E_B$. *Figure 2c* shows that the contours corresponding to $P_{\text{TF}} = 0.9$ at varying values of $E_B$ all take an approximate symmetrical L-shape, thus supporting the design principle $n_b = n_t$. For a given $E_B$, we denote the optimal values as $n_b^* = n_t^*$, and show them as hexagrams in the plot.

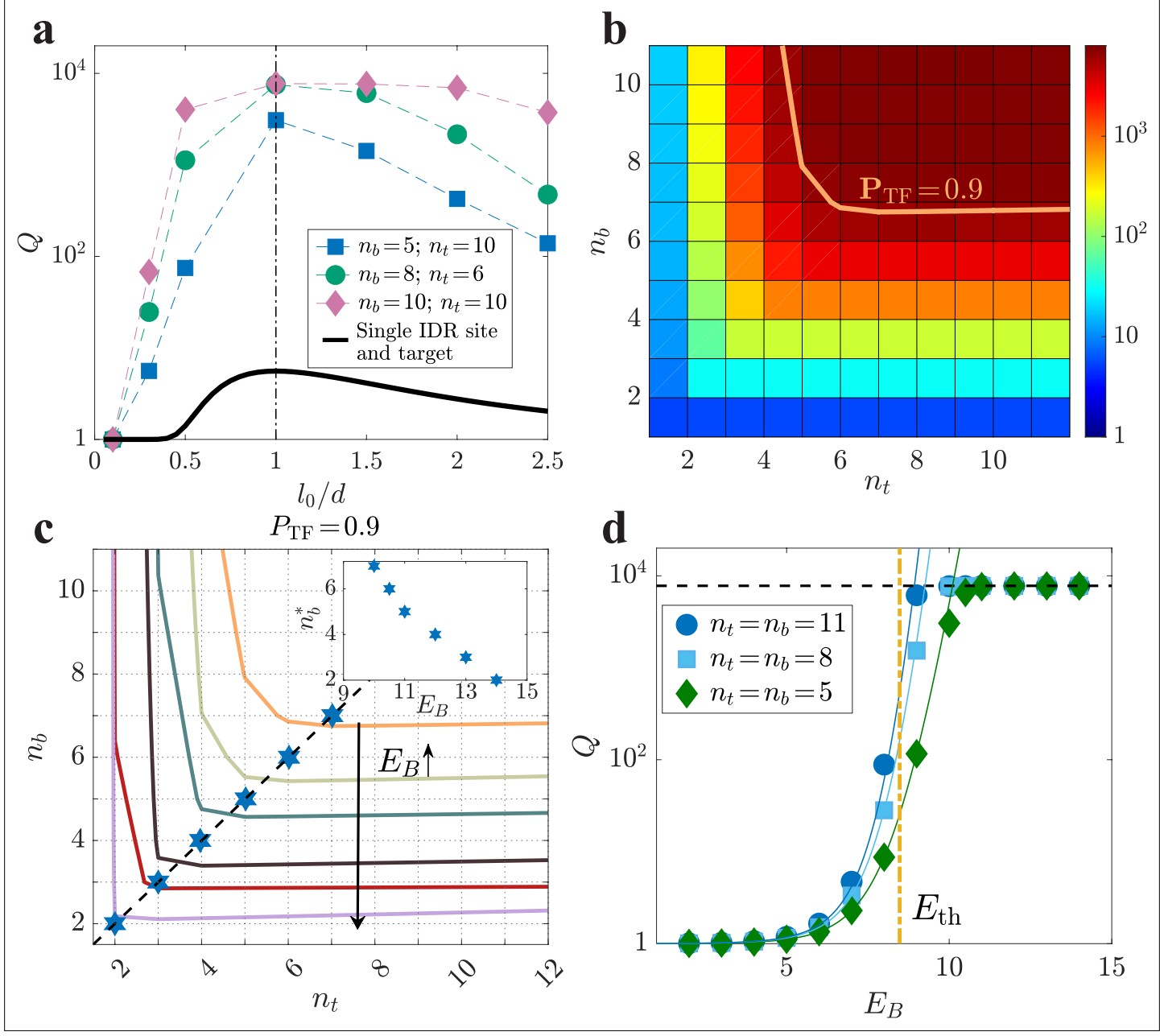

**Figure 2.** Design principles across various parameters: $l_0/d$, $n_b$, $n_t$, and $E_B$. (**a**) Plots of $Q \equiv P_{\mathrm{TF}}/P_{\mathrm{simple}}$ varying with $l_0/d$ are shown for several combinations of $n_t$ and $n_b$ for typical values of the relevant quantities: the nucleus volume is $1\mu\mathrm{m}^3$, $E_{\mathrm{DBD}}=15$, $E_B=10$, $V_1=(0.34\mathrm{nm})^3$, and $d=\sqrt{50}\cdot 0.34\mathrm{nm}$. (**b**) A heatmap of $Q$ vs. $n_b$ and $n_t$ at $E_B=10$ and $d=l_0$. (**c**) Contour lines at different $E_B$ at $P_{\mathrm{TF}}=0.9$. Blue hexagrams represent the design principle $n_b^*=n_t^*$, with the black dashed line as a visual guide. The inset shows the dependence of $n_b^*$ on energy. (**d**) $Q$ for varying $E_B$ at $d=l_0$ for several $n_b=n_t$. The vertical line represents $E_{\mathrm{th}}$ as determined by *Equation 4*, the solid curves illustrate the approximation of $Q$ obtained from *Equation 3*, and the horizontal line indicates where $P_{\mathrm{TF}}=1$.

We further find that $n_b^*$ decreases with increasing $E_B$, as displayed in the inset. These relationships also hold at $P_{\mathrm{TF}}=0.5$ (Appendix 5. Robustness of the relationships: $n_b^* \sim n_t^*$ and $n_b^*$ decreases with increasing $E_B$).

We examine the variation of $Q$ with respect to $E_B$ while keeping $n_b=n_t$ constant. In *Figure 2d*, as the binding energy $E_B$ increases, $Q$ increases from a value of $\sim 1$ to its saturation value, $1/P_{\mathrm{simple}}$, in a switch-like manner: that is, there is a characteristic energy scale, $E_{\mathrm{th}}$. To estimate $E_{\mathrm{th}}$, when $Q \ll 1/P_{\mathrm{simple}}$ (i.e. $P_{\mathrm{TF}} \ll 1$), we approximate the expression of $Q$ as:

$$Q \approx 1 + \Sigma_{n=1}^{n_b} \Sigma_{\mathrm{orn}} \prod_{i=1}^{n} \mathcal{P}(E_B, r_{\mathrm{orn},i}, l_{\mathrm{orn},i}), \tag{3}$$

where $l_{\mathrm{orn},i}$ and $r_{\mathrm{orn},i}$ denote the segment length between two adjacent bound sites and the distance associated with their respective targets, respectively, and 'orn' represents the orientation of the given configuration (Appendix 2. Derivation of the binding probability). *Equation 3* reduces to *Equation 1* when $n_t = n_b = 1$. To substantially increase $Q$, the configuration that contributes the most to $Q$, would satisfy $(\mathcal{P}(E_B, d, d))^{n_b} \gg 1$. Thus, the characteristic energy $E_{\mathrm{th}}$ could be estimated using the formula $\mathcal{P}(E_{\mathrm{th}}, d, d) = 1$, resulting in:

$$E_{\mathrm{th}} = \frac{3}{2}\left[\ln(\frac{2\pi}{3\phi^2}) + 1\right], \tag{4}$$

where $\phi = V_1^{1/3}/d$ represents the proportion of IDR targets within the antenna. *Equation 4* is a direct consequence of the energy-entropy trade-off. Increasing $\phi$ leads to a decrease in entropy loss due to binding the IDR, subsequently reducing the demands on binding energy. Given that $\phi^2 = 0.02$, $E_{\mathrm{th}}$ is about $8.5k_BT$. The vertical dashed line in *Figure 2d*, obtained from *Equation 4*, effectively captures the region of significant increase for large values of $n_b$. Hence, we obtain the third design principle: $E_B > E_{\mathrm{th}}$, which ensures the significant binding advantage of TFs with IDRs.

So far, we have found that the binding design principles for TFs with IDRs are as follows: (i) target distance $d$ should be as small as possible, (ii) IDR segment binding distance $l_0$ should be comparable to $d$, (iii) it is preferable to have fewer binding sites $n_b^*$ and binding targets $n_t^*$ as the binding strength $E_B$ increases, with $n_b^* \sim n_t^*$, and (iv) should be larger than a threshold that solely depends on the proportion $\phi$ of IDR targets within the antenna, as indicated by *Equation 4*. In our estimation, with $\phi^2 = 0.02$, we obtain $E_{\mathrm{th}} \approx 8.5k_BT$. The above principles have also been validated in scenarios involving Poisson statistics of binding site distances and target distances, thus indicating their robust applicability (*Appendix 6—figure 1*).

## Search time of the TF from the dynamics

In this section, we study the design principles of the TF *search time*, a dynamic process. Here, we confine the TF with $\tilde{n} \equiv n_b + 1$ sites ($n_b$ IDR sites and one DBD site) to a sphere of radius $R$, the cell nucleus, and use over-damped coarse-grained molecular dynamics (MD) simulations. The sites and targets interact via a short-range interaction. The DBD target with a radius $a$ is placed at the sphere's center. See the sketch in *Figure 1a*. Once the DBD finds its target, the search ends. We denote the mean total search time (i.e. the mean first passage time, MFPT), as $t_{\mathrm{total}}$. We estimate it by averaging 500 simulations, each starting from an equilibrium configuration of the TF at a random position in the nucleus. Simulation details are provided in the Materials and Methods section.

By conducting comprehensive simulations of $t_{\mathrm{total}}$ across various parameter values of $L$, $E_B$, and $\tilde{n}$, we were able to identify its minimum value, which is approximately at $L^* = 300$bp, $E_B^* = 11$, and $\tilde{n}^* = 4$ for $R = 500$bp $\approx 1/6\mu$m. Here, one base pair (bp) is equal to $0.34$nm. To speed up the simulations, we utilized several computational tricks, including adaptive time steps and treating the TF as a point particle when it is far from the antenna (see Materials and methods). Despite this, the simulations require $10^5$ CPU hours due to the IDR trapped in the binding targets on the antenna, which results in the slow dynamics as found in the previous simulation works (*Vuzman and Levy, 2010*; *Marcovitz and Levy, 2013*). The probability distribution function (pdf) of individual search times exponentially decays with a typical time equal to $t_{\mathrm{total}}$, see inset of *Figure 3*. This indicates that $t_{\mathrm{total}}$ provides a good characterization of the search time.

We find that $t_{\mathrm{total}}$ varies non-monotonically with $L$ at fixed $E_B^*$ and $\tilde{n}^*$ as shown in *Figure 3*. Once a TF attaches to one of its targets, it will most likely diffuse along the antenna until reaching the DBD target (Appendix 7. TF walking along the antenna once it attaches, and $t_{\mathrm{total}}$ vs. $E_B$ and vs. $\tilde{n}$). It means that the optimal search process is a single round of 3D diffusion followed by 1D diffusion along the antenna. In Appendix 8, a single round of 3D-1D search is typically the case in the optimal search process., by analytically calculating the mean total search time using a coarse-grained model, we find that a single round of 3D-1D search is typically the case in the optimal search process. Note that our model differs profoundly from the conventional facilitated diffusion (FD) model in bacteria: In the latter, the TF is assumed to bind non-specifically to the DNA. Within our model, the antenna

**Figure 3.** Mean search time varying with the antenna length , $t_{3D}$ and $t_{1D}$ vs. at $E_B^*$ and $\tilde{n}^*$. The solid curves correspond to $t_{total}$ , $t_{3D}$ and $t_{1D}$ in *Equation 5*. Inset: probability density function (pdf) exponentially decays at large $t$.pdf $\approx \exp\left(-t/t_{total}\right)/t_{total}$ (black solid line). The unit $t_0$ represents the time for one AA to diffuse 1bp. $a = 0.5$bp, $d = 10$bp, and $l_0 = 5$bp.

length $L$ can be tuned by changing the placement of the IDR targets on the DNA, making it an adjustable parameter rather than the entire genome length. As opposed to the FD model, which requires numerous rounds of DNA binding-unbinding to achieve minimal search time, within our model, the minimal search time is achieved for a single round of 3D and 1D search.

Interestingly, when we observe the motion of a TF along the antenna, IDR sites behave like 'tentacles', as illustrated in *Figure 1c*, constantly binding and unbinding from the DNA. We refer to this 1D walk as 'octopusing', inspired by the famous 'reptation' motion in entangled polymers suggested by de Gennes (*de Gennes and Leger, 1982*; see *Video 1*). We note that previous numerical studies *Vuzman and Levy, 2010*; *Vuzman and Levy, 2012* used comprehensive, coarse-grained, MD simulations to demonstrate that including heterogeneously distributed binding sites on the

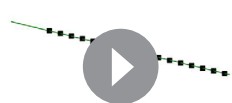

**Video 1.** Video of the 1D octopusing walk simulation.
https://elifesciences.org/articles/104956/figures#video1

IDR could facilitate intersegmental transfer between different regions of the DNA, which is reminiscent of our octopusing scenario.

The mean 3D search time, $t_{3D}$, and the mean 1D search time, $t_{1D}$ are also shown in **Figure 3**. To evaluate $t_{3D}$, we take advantage of several insights: first, if the TF gyration radius, $r_p = \sqrt{\tilde{n}/6}l_0$, is comparable or larger than the distance between targets, $d$, we may consider the chain of targets as, effectively, a continuous antenna. Making the targets denser would hardly accelerate the search time, reminiscent of the well-known chemoreception problem where high efficiency is achieved even with low receptor coverage (**Berg and Purcell, 1977**). Second, when the distance between the TF and the antenna is smaller than $r_p$, one of the binding sites on the TF would almost surely attach to the targets. This allows us to approximately map the problem to the MFPT of a particle hitting an extremely thin ellipsoid with long axis of length $L$ and radius $r_p$, the solution of which is $\ln(L/r_p)/L$ (**Berg, 2018**). We find that this scaling captures well the dependence of $t_{3D}$ on the parameters, with a prefactor $\alpha \sim O(1)$.

For the 1D search from a single particle picture, since the TF is unlikely to detach once it attaches, it performs a 1D random walk. So, taken together, the mean total search time is

$$t_{\text{total}} = t_{3D} + t_{1D} \approx \alpha \frac{\tilde{n}R^3}{DL} \ln(L/r_p) + \beta \frac{L^2}{D}, \tag{5}$$

where $D$ is the 3D diffusion coefficient of one site. For a TF with $\tilde{n}$ sites, its 3D diffusion coefficient, $D_3$, is $D/\tilde{n}$ and $\beta$ are fitting parameters.

The semi-analytical formulas of $t_{1D}$, $t_{3D}$, and $t_{\text{total}}$, denoted by the solid lines in **Figure 3**, agree well with the simulation results for $\alpha = 0.5$ and $\beta = 14.5$.

The theoretical result for the time spent in 3D was originally calculated for a point searcher (**Berg, 2018**). In that case, a value of $\alpha = 2/3$ was obtained. $\beta$ is larger at larger $E_B$ (see Appendix 7. TF walking along the antenna once it attaches, and $t_{\text{total}}$ vs. $E_B$ and vs. $\tilde{n}$).

We also examine the dependence of $t_{\text{total}}$ on the key parameters $E_B$ and $\tilde{n}$, in the vicinity of the optimal parameters which minimize the total search time (Appendix 7. TF walking along the antenna once it attaches, and $t_{\text{total}}$ vs. $E_B$ and vs. $\tilde{n}$). We find that the affinity of TFs to the antenna decreases rapidly with decreasing the IDR length (**Brodsky et al., 2020**) and $E_B$. Consequently, when $\tilde{n} < \tilde{n}^*$ or $E_B < E_B^*$, multiple search rounds are needed to reach the DBD target, resulting in a significant escalation of $t_{\text{total}}$. When $\tilde{n} > \tilde{n}^*$, $t_{\text{total}}$, $t_{3D}$, and $t_{1D}$ agree with **Equation 5**. When $E_B > E_B^*$, since $t_{3D}$ is independent of $E_B$, the variation in $t_{\text{toS7tal}}$ stems from $t_{1D}$ which exponentially increases with $E_B$.

The minimal search time for a simple TF is $t_{\text{simple}} = R^3/(3Da)$ (**Berg, 2018**).

For the TF with an IDR, by minimizing the expression in **Equation 5**, we find that the optimal length approximately scales as $L^* \propto R$, and therefore the minimal search time, $t_{\text{min}}$, is approximately proportional to $R^2$.

At $R = 500\,\text{bp} \approx 1/6\mu m$, the search time is more than ten times shorter than a simple TF, despite the latter's diffusion coefficient being several times larger. Based on our analytic result, we expect $\sim 50$ fold enhancement for the experimental parameters of yeast note that we assume the scaling $t_{\text{min}} \sim R^2$ still holds in this regime; this formula is approximate since it does not take into account the persistence length of DNA (**Guilbaud et al., 2019**), which is several-fold shorter than the optimal antenna length $L^*$. We have verified that reducing the DNA persistence length, which promotes increased DNA coiling, results in only a modest increase in mean search time. Even under extreme coiling conditions, the increase remains below 30% of the baseline value, as detailed in Appendix 9. Robustness of mean total search time with changing the antenna geometry. Converting $t_{\text{min}}$ into a second-order on-rate yields approximately $10^8 - 10^9\,\text{M}^{-1}s^{-1}$, assuming $D \approx 10^{-13}m^2/s$ and a TF copy number of 100. We also verified that the 3D diffusion coefficient slightly changes under a complex DNA configuration, where a point-searcher TF non-specifically interacts with the entire DNA, indicating the robustness of our optimal search process (see Appendix 10. MSD in a complex DNA configuration).

Next, we characterize the dynamics of the octopusing walk (Appendix 11. Properties of the octopusing walk). During this process, different IDR binding sites constantly bind and unbind from the antenna, leading to an effective 1D diffusion. A *priori*, one may think that the TF faces conflicting demands: it has to rapidly diffuse along the DNA while also maintaining strong enough binding to prevent detaching from the DNA too soon. Indeed, within the context of TF search in bacteria, this is known as the speed-stability paradox (**Slutsky and Mirny, 2004**). Importantly, the simple picture of

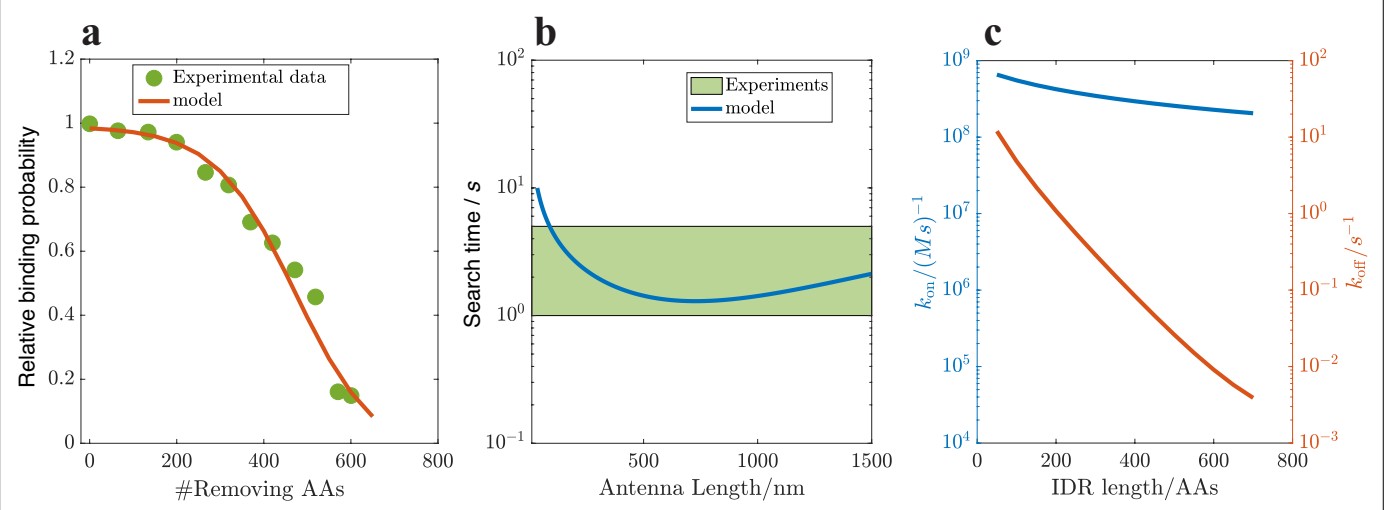

**Figure 4.** Comparisons with experimental results. (**a**) Relative binding probability varies with the truncation length of the IDR, which quantitatively agrees with the experimental data in **Brodsky et al., 2020**, where we only vary $E_{DBD}$ (binding energy of DBD) to ensure that the maximal value reaches 1 at zero truncation length. Antenna length $L = 1000$nm, $E_{DBD} = 22$ and $E_B = 11$. Other parameters are set the same as in **Figure 2**. (**b**) The search time estimated by our model (from **Equation 5** and divided by a TF copy number of 100) is quantitatively comparable with the experimental results (**Larson et al., 2011**), as guided by the shaded area. c, The on-rate $k_{on} = t_{total}/V_c$ and off-rate $k_{off} = (1 - P_{TF})/(P_{TF}t_{total})$ varying with the IDR length, indicating that $k_D \equiv k_{off}/k_{on}$ ranges from 0.01 to 10 nM, with its variation primarily dominated by the off-rate.

the octopusing walk explains an elegant mechanism to bypass this paradox, achieving a high diffusion coefficient with a low rate of detaching from the DNA. This comes about due to the compensatory nature of the dynamics: there is, in effect, a feedback mechanism whereby if only a few sites are bound to the DNA, the on-rate increases, and similarly, if too many are bound (thus prohibiting the 1D diffusion) the off-rate increases. Remarkably, this mechanism remains highly effective even when the typical number of binding sites is as low as two. Further details are provided in Appendix 11. Properties of the octopusing walk.

## Comparing with experimental results

In **Brodsky et al., 2020**, the binding probability of the TF Msn2 in yeast was measured, where the native IDR was shortened by truncating an increasingly large portion of it. The results indicate that binding probability decreases with increasing truncation of the IDR, as illustrated in **Figure 4a**. We utilized our model to calculate the relative binding probability compared to $P_{TF}$ at a zero truncating length. Using the DBD binding energy $E_{DBD}$ as our only fitting parameter, our model achieves quantitative agreement with the experimental data. In **Figure 4b**, we estimate the search time as a function of $L$ for $R = 1\mu m$ and $D_0 = 10^{-10} m^2/s$ for a single AA. The estimated timescales are comparable to empirical results, which are 1 to 10 seconds (**Kumar et al., 2023**; **Larson et al., 2011**), as indicated by the shaded area. We have also estimated the on-rate $k_{on}$ and off-rate $k_{off}$ of the TF as they vary with IDR length, as shown in **Figure 4c**. In the kinetic proofreading mechanism (**Hopfield, 1974**), specificity is associated with the off-rate, but recent studies (**Marklund et al., 2022**) suggest that in other scenarios, specificity is encoded in the on-rate. Therefore, it is important to explore how specificity is encoded within this mechanism. Our results indicate that the variation in the dissociation constant $k_{off}/k_{on}$ is dominated by $k_{off}$, suggesting that specificity is encoded in the off-rate. The magnitudes and variations of these rates could be directly compared to experimental results in future studies.

## Discussion

In summary, we addressed the design principles of TFs containing IDRs, with respect to binding affinity and search time. We find that both characteristics are significantly enhanced compared to simple TFs lacking IDRs.

The simplified model we presented provides general design principles for the search process and testable predictions. One possible experiment that could shed light on one of our key assumptions is 'scrambling' the DNA (randomizing the order of nucleotides). Our results depend greatly on our assumption that there exists a region on the DNA, close to the DBD target, that contains targets for the IDR binding sites. It is possible to shuffle, or even remove, regions of different lengths around the DBD target and to measure the resulting changes in binding affinity. According to our results, we would expect that shuffling a region smaller in length than the IDR should have minimal effect on the binding affinity and search time, but that shuffling a region longer than the IDR should have a noticeable effect.

Another useful test of our model would be to probe the *dynamics* of the TF search process by varying the waiting time before TF-DNA interactions are measured: Within the ChEC-seq technique (*Zentner et al., 2015*) employed in *Brodsky et al., 2020*, an enzyme capable of cutting the DNA is attached to the TF and activated at will (using an external calcium signal). Sequencing then informs us of the places on the DNA at which the TF was bound at the point of activation. Measuring the binding probabilities at various waiting times — defined as the time between adding calcium ions and adding transcription factors, ranging from seconds to an hour, would inform us both of the TF search dynamics as well as the equilibrium properties: Shorter time scales would capture the dynamics of the search process, while longer time scales would reveal the equilibrium properties of the binding. At short waiting times (within several seconds), we would expect the binding probability to increase with waiting times. By analyzing a normalized binding probability curve as a function of the waiting time $t$, we could fit the curve using the function $1-\exp(-t/t_{\text{search}})$. From this curve, we can infer the search time. Our results suggest that the presence of the IDR in a TF leads to increased binding probabilities over various waiting times and a reduced waiting time to reach a higher saturation binding probability. Furthermore, as shown in *Figure 4* obtained from our model, $k_{\text{on}}$ can be determined from the measurements of $t_{\text{search}}$. Using the binding probability, which is associated with $k_D = k_{\text{off}}/k_{\text{on}}$, $k_{\text{off}}$ can also be calculated.

While our work primarily focused on coarse-grained modeling of the search process and the hypothesized octopusing dynamics, higher-resolution molecular dynamics simulations could provide more detailed molecular insights and further validate our proposed design principles (*Vuzman and Levy, 2010*; *Vuzman and Levy, 2012*). Since these principles are grounded in fundamental physical concepts, such as the energy-entropy trade-off, they are likely to be robust across diverse molecular systems. We anticipate their applicability to other biophysical contexts, such as nonspecific RNA polymerase binding (*Tenenbaum et al., 2023*) and multivalent antibody binding (*Einav et al., 2019*), where the associated dynamic processes would also be of significant interest for future research. Beyond our octopusing mechanism, other mechanisms may also contribute to the search and binding process. IDRs could facilitate complex formation and cooperativity between multiple transcription factors, as suggested by studies showing co-localization at common target promoters (*Morgunova and Taipale, 2017*). IDRs might also enhance efficiency by directing transcription factors to specific nuclear compartments or biomolecular condensates, thereby reducing the effective search space and time (*Sabari et al., 2018*; *Boija et al., 2018*; *Kent et al., 2020*). These complementary mechanisms (*Ferrie et al., 2022*; *Jonas et al., 2025*; *Holehouse and Kragelund, 2024*) suggest that our findings represent one component of a broader regulatory framework governing transcriptional control in eukaryotic cells.

## Materials and methods
### Model

The Rouse Model describes a polymer as a chain of point masses connected by springs and characterized by a statistical segment length $l_0$ resulting from the rigidity of springs $k$ and the thermal noise level $k_B T$, $l_0^2 = 3k_B T/k$ (*Prince, 1953*; *Doi and Edwards, 1988*). We coarse-grain the TF by representing it with a few binding sites for the IDR, and an additional site for the DBD, as illustrated in

*Figure 1b*. The targets are assumed to be positioned at equal spacing, with one target for the DBD and a few targets for the IDR, corresponding to the orange and black squares in *Figure 1c*. We have also verified that relaxing this assumption, and using randomly positioned targets, does not affect our results (Appendix 6. Robustness of the binding design principles). In the Rouse Model, the 3D probability density describing the relative distance in the equilibrium state, denoted as $r$, between any two sites can be derived through Gaussian integration (*Doi and Edwards, 1988*), leading to:

$$f(r, l) = \left(\frac{3}{2\pi l^2}\right)^{\frac{3}{2}} \exp\left(-\frac{3r^2}{2l^2}\right), \tag{6}$$

where $l = \sqrt{n_{\text{seg}}} l_0$ and $n_{\text{seg}}$ is the number of segments between the two sites.

## Dynamical simulations

Each binding site of the TF at position $r_i$ follows over-damped dynamics, which reads:

$$\Delta r_i = \frac{D}{k_B T} f(\{r_i\})\Delta t + \sqrt{2D\Delta t}\, \xi, \tag{7}$$

where the random variable $\xi$ follows the standard normal distribution. $D$ is the diffusion coefficient of the coarse-grained binding site. The force acting on each site is $f(\{r_i\}) \equiv -\partial U_{\text{total}}/\partial r_i$, and

$$U_{\text{total}} = \sum_{i=0}^{n_b-1} \frac{1}{2}k(r_{i+1} - r_i)^2 + \sum_{i,j} E_B \exp\left[-\frac{(r_i - R_j)^2}{2w_d^2}\right], \tag{8}$$

where $R_j$ are the positions of the targets. $w_d = 1\text{bp}$ is the typical width of the short-range interaction between sites and their corresponding targets. The typical distance $l_0$ between coarse-grained binding sites is 5bp. The stiffness of the springs is thus $3k_B T/l_0^2 = 3k_B T/25\text{bp}^2$. To speed up the simulations, when the sites' minimal distance to the antenna is larger than the threshold 10bp, the TF is regarded as a rigid body, and all sites move in sync with their center of mass. We verified that the resulting search time is insensitive to the value of this threshold. When the sites' minimal distance to the antenna is less than 10bp, we consider the excluded volume effect to ensure that different binding sites are not trapped at the same target. Specifically, we use the WCA interaction (*Weeks et al., 1971*):

$$U_{\text{WCA}} = \begin{cases} \dfrac{\sigma^{12}}{|r_i - r_j|^{12}} - \dfrac{2\sigma^6}{|r_i - r_j|^6} + 1, & |r_i - r_j| < \sigma \\ 0, & |r_i - r_j| \geq \sigma \end{cases} \tag{9}$$

where $\sigma = 1\text{bp}$. In this region, we also adopted an adaptive time step: $\Delta t/(\text{bp}^2/D) = \min\left\{1, \frac{k_B T}{\max |f_i| \text{bp}}\right\} \times 0.01$. Note that this choice ensures that the displacement of the sites at every step is much smaller than the binding site size $w_d$. Due to the IDR trapped in the binding targets on the antenna, which results in slow dynamics, the simulation time becomes significantly longer, as found in previous simulation studies (*Vuzman and Levy, 2010*; *Marcovitz and Levy, 2013*). This slowdown is similar to the behavior observed in glass-forming liquids (*Berthier and Biroli, 2011*). As a result, the individual simulation time steps range from $10^9$ to $10^{11}$.

## Acknowledgements

We thank Samuel Safran, David Mukamel, Yariv Kafri, Yaakov Levy, Luyi Qiu, and Zily Burstein for insightful discussions. We thank the Clore Center for Biological Physics for their generous support.

## Additional information

### Competing interests

Ariel Amir: Reviewing editor, eLife. The other authors declare that no competing interests exist.

## Funding
No external funding was received for this work.

## Author contributions
Wencheng Ji, Ori Hachmo, Naama Barkai, Ariel Amir, Investigation, Writing – original draft, Writing – review and editing

## Author ORCIDs
Wencheng Ji ⬤ https://orcid.org/0000-0003-1465-3953
Naama Barkai ⬤ https://orcid.org/0000-0002-2444-6061
Ariel Amir ⬤ https://orcid.org/0000-0003-2611-0139

Reviewer #1 (Public review): https://doi.org/10.7554/eLife.104956.3.sa1
Reviewer #2 (Public review): https://doi.org/10.7554/eLife.104956.3.sa2
Author response https://doi.org/10.7554/eLife.104956.3.sa3

---

# Additional files

## Supplementary files
MDAR checklist

## Data availability
Data and code availability: The data supporting this study's findings and the codes used in this study are available from https://github.com/wenchengJi/code_TF (copy archived at *Ji, 2024*).

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

## Appendix 1

### Broad applicability of probability density $f(r, l)$

We verify the broad applicability of *Equation 6* in the main text, regardless of the specific details of microscopic interactions. To demonstrate this, we only require that the density function of the end-to-end distance $r$ for a segment containing $m$ amino acids (AAs), denoted as $f(r, l_0)$, complies with *Equation 6*. Specifically, it is.

$$f(r, l_0) \approx \left( \frac{3}{2\pi l_0^2} \right)^{3/2} \exp\left( -\frac{3r^2}{2l_0^2} \right), \tag{10}$$

where $l_0$ is the typical length of the segment. When $m$ is large, according to the central limit theorem, $r$ should conform to the distribution in *Equation 10* with a variance of $l_0^2 = m a_{\text{eff}}^2$, where $a_{\text{eff}}$ represents the effective length of an individual AA.

Next, we will consider a specific form of interaction, of springs with a finite rest length. In one limit (where the rest length vanishes), this corresponds to the Rouse model (where *Equation 10* is exact for any value of $m$, not only asymptotically) and in another limit (where the springs are very stiff) this corresponds to a chain of rods. As we shall see, *Equation 10* provides an excellent approximation already for moderate values of $m$.

To proceed, we consider two adjacent AAs at positions $\vec{r}_i$ and $\vec{r}_{i+1}$ with an interaction denoted by $u(|\vec{r}_i - \vec{r}_{i+1}|)$. Their probability density is $\rho(\vec{r}_i, \vec{r}_{i+1}) \propto \exp\left( -u(|\vec{r}_i - \vec{r}_{i+1}|)/(k_B T) \right)$. $a_{\text{eff}}^2$ can be calculated using $\rho(\vec{r}_i, \vec{r}_{i+1})$:

$$a_{\text{eff}}^2 = \int d\vec{r} \int d\vec{r}_i \int d\vec{r}_{i+1} \, |\vec{r}_i - \vec{r}_{i+1}|^2 \rho(\vec{r}_i, \vec{r}_{i+1}) \delta\left( \vec{r} - \left( \vec{r}_i - \vec{r}_{i+1} \right) \right). \tag{11}$$

Specifically, for a spring-like interaction with a stiffness constant $k_0$ and a rest length $b_0$, the interaction is given by $u = \frac{1}{2} k_0 \left( |\vec{r}_i - \vec{r}_{i+1}| - b_0 \right)^2$. This formula can also be interpreted as a harmonic approximation of the interaction around the equilibrium distance $b_0$. In this case, $a_{\text{eff}}^2$ is derived analytically:

$$a_{\text{eff}}^2 = a_0^2 + b_0^2 + \frac{2b_0^2 a_0^2}{a_0^2 + 3b_0^2}(1 + \frac{2a_0^3}{6b_0^2 a_0 + \sqrt{6\pi} b_0 \left( a_0^2 + 3b_0^2 \right) \left( 1 + \text{Erf}(\sqrt{\frac{3}{2}} \frac{b_0}{a_0}) \right) \exp(\frac{3b_0^2}{2a_0^2})}), \tag{12}$$

where $a_0^2 \equiv 3k_0/k_B T$ and $\text{Erf}(z) = \frac{2}{\sqrt{\pi}} \int_0^z \exp(-x^2/2) dx$ is the error function. When $b_0 = 0$, we have $a_{\text{eff}} = a_0$, and *Equation 10* holds for any $m$. When $b_0 \gg a_0$, then $a_{\text{eff}} \approx b_0$ corresponds to a chain of rods where the angle between consecutive rods is free to rotate.

*Appendix 1—figure 1a* shows that the numerical results obtained from this harmonic interaction are in good agreement with *Equation 10* even at moderate values of $m$. The inset displays the variation of $a_{\text{eff}}^2/a_0^2$ with $b_0/a_0$ obtained from *Equation 12*. In *Appendix 1—figure 1b*, with $b_0 = a_0$, $f(r, l_0)$ rapidly converges to *Equation 10*.

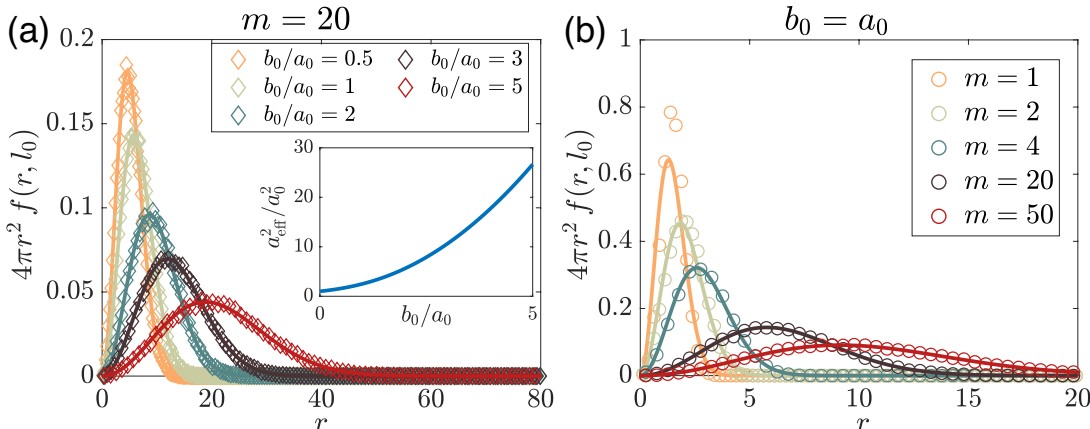

**Appendix 1—figure 1.** $f(r, l_0)$ aligns well with *Equation 10* (solid curves) for moderate $m$ at various. $b_0$ Inset: $a_{\text{eff}}^2/a_0^2$ vs. $b_0/a_0$ from *Equation 12*. (**b**) *Equation 10* serves as a good approximation (solid lines) even at small $m$.

## Appendix 2.

### Derivation of the binding probability

In this section, we derive the generalized binding probability $P_{TF}$ for $n_b$ IDR binding sites and $n_t$ IDR targets. To achieve this generalization, we consider all the possible number of bindings, $n$ from 0 to $\min(n_b, n_t)$, and all possibilities for binding orientations for each $n$. Note that the energy and entropy trade-off is naturally included in our model, and the generalized formula of $P_{TF}$ is also applicable for non-uniformly distributed binding targets and binding sites.

Based on the Rouse model, the system's energy can be expressed as $\frac{k}{2}\sum_{i=1}^{n_b}(\vec{r}_i - \vec{r}_{i-1})^2 = \frac{3k_BT}{2l_0^2}\sum_{i=1}^{n_b}(\vec{r}_i - \vec{r}_{i-1})^2$, where $\vec{r}_i$ represents the position of the binding site $i$. The probability density in the absence of binding reads:

$$\rho(\vec{r}_0, ..., \vec{r}_{n_b}) = \frac{1}{V_c}\left(\frac{3}{2\pi l_0^2}\right)^{\frac{3}{2}n_b} e^{-\frac{3}{2l_0^2}\sum_{i=1}^{n_b}(\vec{r}_i - \vec{r}_{i-1})^2}, \tag{13}$$

where $\int d^3\vec{r}_0... \int d^3\vec{r}_{n_b}\rho(\vec{r}_0, ..., \vec{r}_{n_b}) = 1$.

Now we consider a given configuration with $n$ IDR sites bound, where IDR sites $q_1 < ... < q_n$ are bound to targets $s_1, ..., s_n$. The positions of these bound targets are at $\vec{R}_{s_1}, ..., \vec{R}_{s_n}$. The DBD site is denoted by $q_0 = 0$ and its target is denoted by $s_0 = 0$. There are $n_b - n$ unbound IDR sites, indicated by $\bar{q}_1 < ... < \bar{q}_{n_b-n}$. The corresponding binding probability is given by:

$$P_{\text{bound}}(\text{orn}, n) = \frac{1}{Z}\prod_{i=0}^{n}\int_{\vec{r}_{q_i}\in V_1} d^3\vec{r}_{q_i}\prod_{\ell=1}^{n_b-n}\int_{\vec{r}_{\bar{q}_\ell}\notin V_1} d^3\vec{r}_{\bar{q}_\ell}\, \rho(\vec{r}_0, ..., \vec{r}_{n_b})\, e^{E_{\text{DBD}}+nE_B} \tag{14}$$

where 'orn' stands for the orientation of the given configuration, $Z$ is the normalization factor, and $E_{\text{DBD}}$ is the DBD site binding energy for its target. There are $\binom{n_b}{n}\binom{n_t}{n}n!$ possible binding orientations at $n$. The key insight in evaluating this integral is to realize that we can use the result of *Equation 6* of the main text to integrate out the degrees of freedom associated with the sites between $q_{i-1}$ and $q_i$:

$$\prod_{\bar{q}_\ell=q_{i-1}+1}^{q_i-1}\left[\int d^3\vec{r}_{\bar{q}_\ell}\left(\frac{3}{2\pi l_0^2}\right)^{3/2}\exp\left(-\frac{3(\vec{r}_{\bar{q}_\ell} - \vec{r}_{\bar{q}_\ell-1})^2}{2l_0^2}\right)\right] = f\left(|\vec{r}_{q_i} - \vec{r}_{q_{i-1}}|, \sqrt{q_i - q_{i-1}}l_0\right). \tag{15}$$

Since $r_{q_{i-1}}$ and $r_{q_i}$ are confined to a very small volume $V_1$, to an excellent approximation we may simply evaluate $f\left(|\vec{r}_{q_i} - \vec{r}_{q_{i-1}}|, \sqrt{q_i-q_{i-1}}l_0\right)$ in *Equation 15* at $\vec{r}_{q_{i-1}} = \vec{R}_{s_{i-1}}$ and $\vec{r}_{q_i} = \vec{R}_{s_i}$, to obtain:

$$P_{\text{bound}}(\text{orn}, n) \simeq \frac{1}{Z}\frac{V_1^{n+1}}{V_c}\prod_{i=1}^{n}f\left(|s_i - s_{i-1}|d, \sqrt{q_i - q_{i-1}}l_0\right)e^{E_{\text{DBD}}+nE_B} = \frac{1}{Z}r_{\text{DBD}}\prod_{i=1}^{n}\mathcal{P}(E_B, r_{\text{orn},i}, l_{\text{orn},i}), \tag{16}$$

where $r_{\text{DBD}} \equiv \frac{V_1}{V_c}e^{E_{DBD}}$, $r_{\text{orn},i} = |s_i - s_{i-1}|d$, $l_{\text{orn},i} = \sqrt{q_i - q_{i-1}}l_0$, and $\mathcal{P}(E_B, r_{\text{orn},i}) = e^{E_B}V_1 f(r_{\text{orn},i}, l_{\text{orn},i})$.

To give a specific example, consider the binding topology corresponding to *Appendix 2—figure 1*. In this case, $n = 3$, $l_{\text{orn},1} = l_0$ (since there is one spring between the DBD and the first binding site), $l_{\text{orn},2} = \sqrt{3}l_0$ (since there are three springs between $q_1$ and $q_2$) and similarly $l_{\text{orn},3} = l_0$. *Equation 15* implies that the contribution of this diagram to the binding probability is: $\frac{1}{Z}\frac{V_1^4}{V_c}\frac{27}{(2\pi l_0^2)^{9/2}}e^{3E_B+E_{\text{DBD}}-5d^2/l_0^2}$.

Note that while in this example, the order of the target sites is monotonic (i.e. as we increase the index of the binding sites, we progressively move further away from the DBD), in principle, we should sum over *all* diagrams, including those where the order is non-monotonic. At $n = 0$, $P_{\text{bound}}(\text{orn}, 0) = \frac{1}{Z}r_{\text{DBD}}$.

**Appendix 2—figure 1.** A diagram of the bound configurations at $n=3$.

Similarly, for the state where the DBD is unbound and $n \geq 2$ IDR sites are bound, the binding probability reads:

$$P_{\text{unbound}}(\text{orn}, n) = \frac{1}{Z} \int_{\vec{r}_0 \notin V_1} d^3\vec{r}_0 \prod_{\ell=1}^{n} \int_{\vec{r}_{q\ell} \in V_1} d^3\vec{r}_{q\ell} \prod_{\ell=1}^{n_b-n} \int_{\vec{r}_{\bar{q}\ell} \notin V_1} d^3\vec{r}_{\bar{q}\ell} \rho(\vec{r}_0, ..., \vec{r}_{n_b}) e^{nE_B} \tag{17}$$

$$\simeq \frac{1}{Z} r_{\text{IDR}} \prod_{i=2}^{n} \mathcal{P}(E_B, r_{\text{orn},i}, l_{\text{orn},i}), \tag{18}$$

where $r_{\text{IDR}} \equiv \frac{V_1}{V_c} e^{E_B}$. For $n=1$, $P_{\text{unbound}}(\text{orn}, 1) = \frac{n_t n_b}{Z} r_{\text{IDR}}$. Here $n_t n_b$ represents the number of possibilities for one of the $n_b$ IDR sites bound to one of the $n_t$ targets. For $n=0$, $P_{\text{unbound}}(\text{orn}, 0) = \frac{1}{Z} \prod_{i=0}^{n_b} \int_{\vec{r}_i \notin V_1} d^3\vec{r}_i \rho(\vec{r}_0, ..., \vec{r}_{n_b}) \simeq \frac{1}{Z}$.

Since the sum of all the possibilities is equal to 1, as given by: $\sum_{p=0}^{n_{\text{min}}} \sum_{\text{orn}} (P_{\text{bound}} + P_{\text{unbound}}) = 1$, where $n_{\text{min}} = \min\{n_b, n_t\}$, we can determine the normalization factor. Therefore, the binding probability $P_{\text{TF}}$ for the DBD bound in its target can be expressed as:

$$P_{\text{TF}} = \frac{r_{\text{DBD}} \left(1 + \sum_{n=1}^{n_{\text{min}}} \sum_{\text{orn}} \prod_{i=1}^{n} \mathcal{P}(E_B, r_{\text{orn},i}, l_{\text{orn},i})\right)}{r_{\text{DBD}} \left(1 + \sum_{n=1}^{n_{\text{min}}} \sum_{\text{orn}} \prod_{i=1}^{n} \mathcal{P}(E_B, r_{\text{orn},i}, l_{\text{orn},i})\right) + 1 + r_{\text{IDR}} \left(n_t n_b + \sum_{n=2}^{n_{\text{min}}} \sum_{\text{orn}} \prod_{i=2}^{n} \mathcal{P}(E_B, r_{\text{orn},i}, l_{\text{orn},i})\right)}. \tag{19}$$

When $P_{\text{TF}} \ll 1$, we can approximate $P_{\text{TF}}$ as

$$P_{\text{TF}} \approx \left(1 + \sum_{n=1}^{n_{\text{min}}} \sum_{\text{orn}} \prod_{i=1}^{n} \mathcal{P}(E_B, r_{\text{orn},i}, l_{\text{orn},i})\right) P_{\text{simple}}, \tag{20}$$

where we have considered the facts that $r_{\text{IDR}} < r_{\text{DBD}}$ because $E_B < E_{\text{DBD}}$, and $P_{\text{simple}} = r_{\text{DBD}}/(1 + r_{\text{DBD}}) \approx r_{\text{DBD}}$ due to $r_{\text{DBD}} \ll 1$. For instance, taking $E_{\text{DBD}} = 15$, $V_1 = 1\text{bp}^3 \approx (0.34\text{nm})^3$, and $V_c = 1\mu m^3$, we find that $r_{\text{DBD}} \sim 10^{-4}$.

At $n_b = n_t = 1$, the generalized **Equation 19** simplifies to the following expression:

$$P_{\text{TF}} = \frac{r_{\text{DBD}} \left[1 + e^{E_B} V_1 f(d, l_0)\right]}{r_{\text{DBD}} \left[1 + e^{E_B} V_1 f(d, l_0)\right] + r_{\text{IDR}} + 1}. \tag{21}$$

Therefore, $P_{\text{TF}}$ can be approximated as $(1 + \mathcal{P}) P_{\text{simple}}$, as illustrated in **Equation 1** of the main text. This approximation is also derivable from **Equation 20**.

# Appendix 3

## $Q$ varying with $l_0/d$ for all combinations of $n_b$ and $n_t$

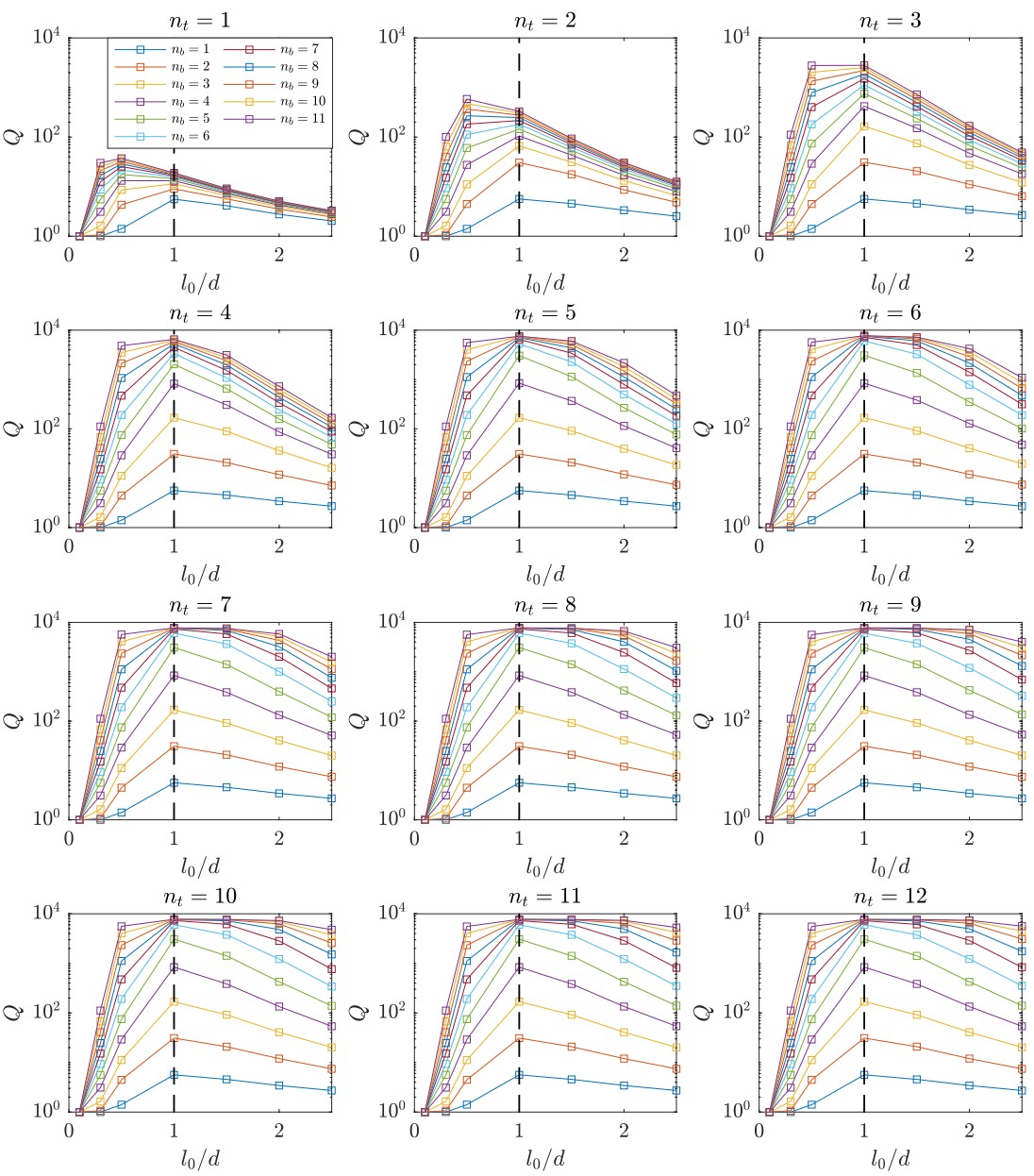

**Appendix 3—figure 1.** Maximal $Q$ is observed at $l_0 \sim d$ for all combinations of $n_b$ and $n_t$.

## Appendix 4

### $P_{\text{TF}}$ **varying with** $n_b$ **and** $n_t$

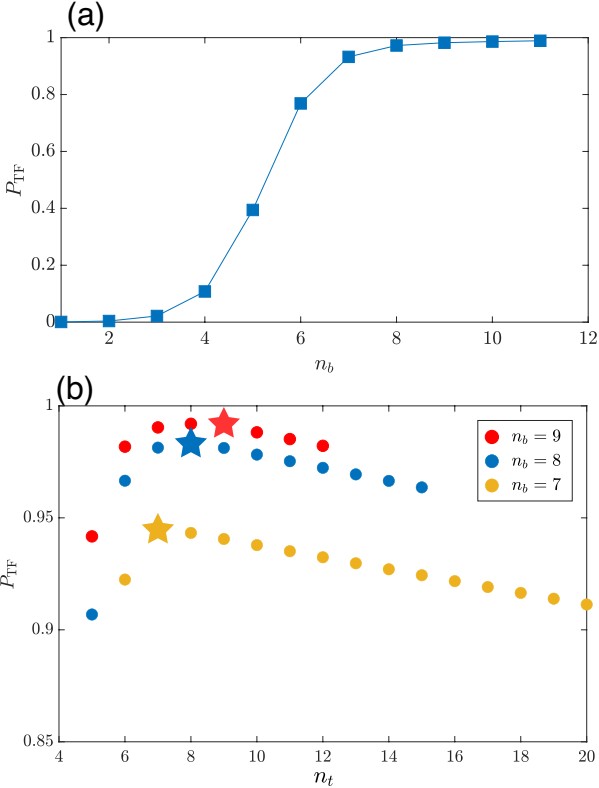

**Appendix 4—figure 1.** Binding probability varying with $n_b$ and $n_1$. (**a**) $P_{\text{TF}}$ varying with $n_b$ at $n_t = 12$. $P_{TF}$ initially increases rapidly, until eventually saturating. (**b**) $P_{\text{TF}}$ varies non-monotonically with $n_t$. $P_{TF}$ slightly decreases with increasing $n_t$ at $n_t > n_b$. Three examples at $n_b = 7, 8, 9$ are displayed. $P_{\text{TF}}$ is denoted by pentagrams at $n_t = n_b$.

## Appendix 5

### Robustness of the relationships: $n_b^* \sim n_t^*$ and $n_b^*$ decreases with increasing $E_B$

*Appendix 5—figure 1* shows the contour lines at $P_{\text{TF}}=0.9$ and $P_{\text{TF}}=0.5$ for various $E_B$. All take an approximate symmetrical L-shape, indicating that for a given $E_B$, the optimal values $n_b^*$ and $n_t^*$ are comparable, as guided by the black dashed line $y=x$. As $E_B$ increases, the contour lines shift towards lower values of $n_b$ and $n_t$, indicating a decrease in $n_b^*$ as $E_B$ increases.

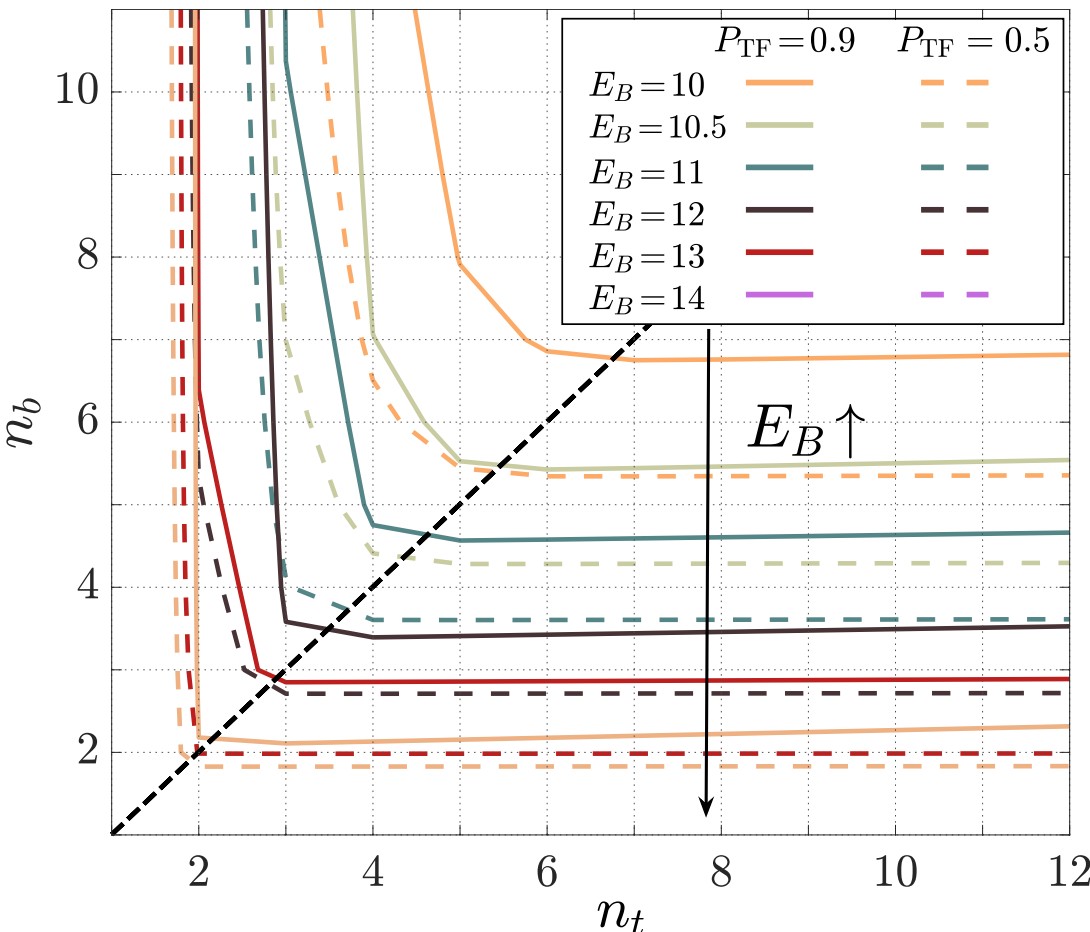

**Appendix 5—figure 1.** Contour lines at $P_{\text{TF}}=0.9$ and $P_{\text{TF}}=0.5$ on the plane defined by the number of binding sites, $n_b$, and the number of binding targets, $n_t$. The black dashed line is added to guide the relationship $n_b = n_t$.

## Appendix 6

### Robustness of the binding design principles

In the main text, we employed uniform binding site distances in TFs, represented as $l_0$, and uniform binding target distances on the DNA, denoted as $d$, to obtain the design principles. Here, we also explore the scenario where both IDR binding sites and their targets are distributed as a Poisson process (i.e. the distance between neighboring sites is exponentially distributed), with mean values of $l_0$ and $d$, respectively. *Appendix 6—figure 1* is similar to *Figure 2* in the main text. We observe that the design principles identified in the main text are applicable under these conditions as well.

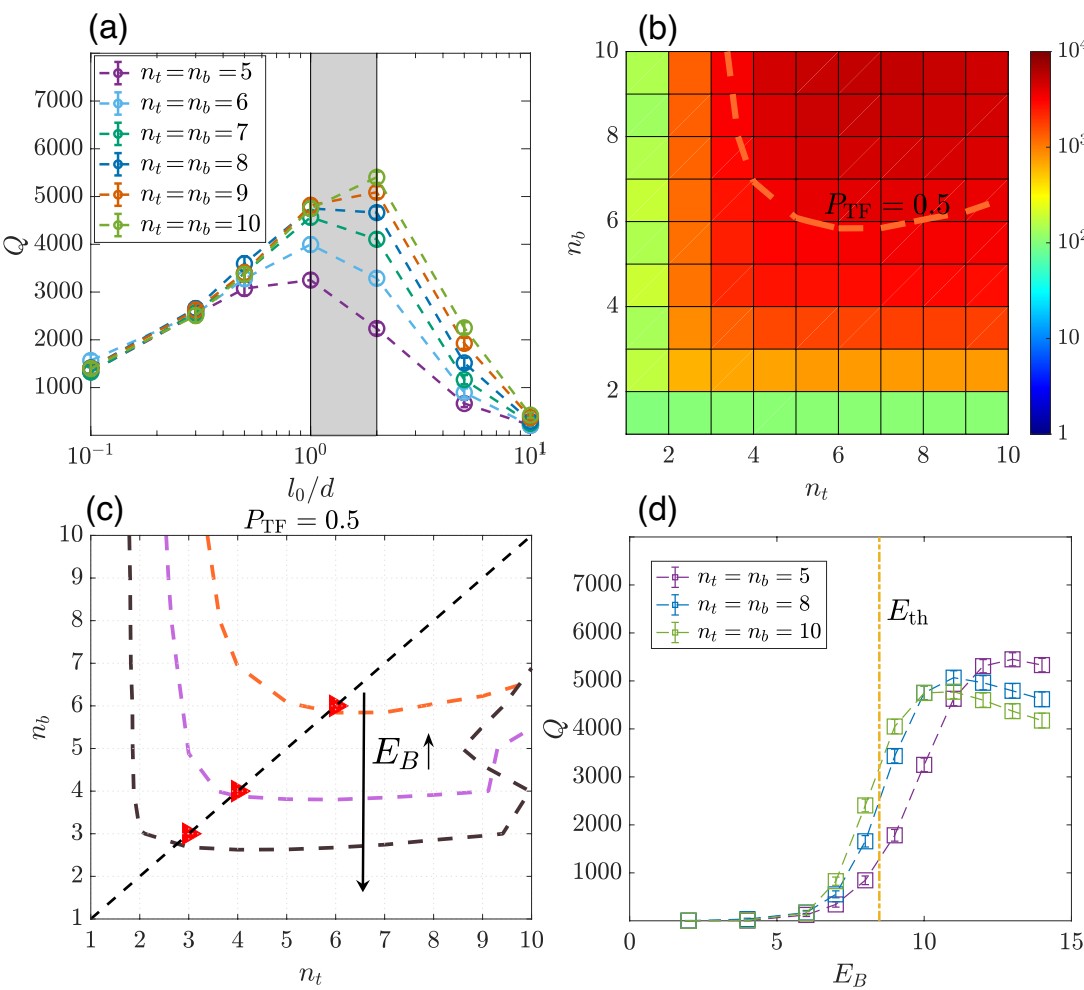

**Appendix 6—figure 1.** Plots of the affinity enhancement, quantified by the ratio $Q \equiv P_{\mathrm{TF}}/P_{\mathrm{simple}}$, for the case where IDR sites as well as their targets are randomly positioned on the IDR and DNA, respectively. See the analogous *Figure 2* in the main text for the case of homogeneously distributed IDR sites and targets. (**a**) $Q$ is maximal when $l_0 \sim d$ (shaded region) for different values of $n_t = n_b$. The average distance between IDR sites is $l_0$, and the average distance between IDR targets is $d$. (**b**) A heatmap of $Q$ at $E_B = 10$ and $l_0 = d$. (**c**) Contour lines at different $E_B$ at $P_{\mathrm{TF}} = 0.5$ exhibit an approximate symmetrical L-shape, which indicates $n_b^* = n_t^*$, as shown by the red triangles. $n_b^*$ decreases with increasing $E_B$. (**d**) $Q$ varies with $E_B$ for several different constants of $n_t = n_b$. The characteristic energy $E_{\mathrm{th}}$, as predicted by *Equation 4* in the main text, corresponds to the region where a notable increase is observed.

# Appendix 7

## TF walking along the antenna once it attaches, and $t_{\text{total}}$ vs. $E_B$ and vs. $\tilde{n}$

We show a typical example of the trajectory of a TF (red) in the cell nucleus (light blue) in *Appendix 7—figure 1a* where the TF walks along the antenna to the DBD target once it attaches the IDR targets at $\tilde{n}^*$ and $E_B^*$. The DBD target is at the origin. In *Appendix 7—figure 1b*, we find that the average number of rounds the TF hits the antenna, $n_{\text{hits}}$, is close to 1, so it is fair to describe the overall search process as two distinct steps with the first step being purely a 3D search and the second a purely 1D search. This is expressed by *Equation 5* in the main text.

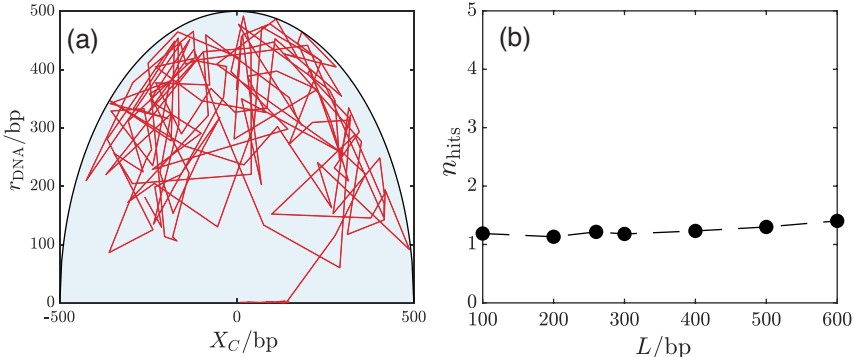

**Appendix 7—figure 1.** Demonstration of a single round of 3D-1D search in simulations. (**a**) A trajectory (red) of TF in the cell nucleus (light blue). is the average distance of the TF to the antenna, and $X_c$ is the position of the center of mass along the antenna direction. The trajectory is displayed with a time interval of $10^6 t_0$. (**b**) The average number of rounds the TF hits the antenna, $n_{\text{hits}}$, before reaching the DBD target. The optimal mean total search time is obtained at $L^* = 300\text{bp}$.

The minimal search time, $t_{\text{min}}$, is obtained at $L^*$, $E_B^*$, and $\tilde{n}^*$. *Appendix 7—figure 2a* shows $t_{\text{total}}$ vs. $E_B$ at fixed $L^*$ and $\tilde{n}^*$. When $E_B > E_B^*$, $t_{\text{3D}}$ does not depend on $E_B$ as we predict (red solid line) and $t_{\text{1D}}$ exponentially increases with increasing $E_B$. The solid blue line shows the exponential dependence of $t_{\text{1D}}$ on $E_B$, also shown in the inset. When $E_B < E_B^*$, the number of rounds of hits on the antenna, $n_{\text{hits}}$, is larger than 1, as displayed in the inset. In this scenario, *Equation 5* in the main text becomes ineffective, resulting in a significantly longer search time than $t_{\text{min}}$.

*Appendix 7—figure 2b* shows $t_{\text{total}}$ vs. $\tilde{n}$ at fixed $L^*$ and $E_B^*$. When $\tilde{n} > \tilde{n}^*$, $t_{\text{3D}}$ and $t_{\text{1D}}$ agree with the prediction of *Equation 5* (red curve and blue curve). Interestingly, we find that $t_{\text{1D}}$ only weakly depends on $\tilde{n}$. This arises due to the compensating effects of attaching and detaching: As $\tilde{n}$ increases, the TF attaches to new targets more easily, speeding up 1D motion. However, it also makes detaching from the current site more difficult, thereby slowing down 1D motion.

When $\tilde{n} < \tilde{n}^*$, *Equation 5* breaks down because of multiple rounds of hits, which is verified in the inset. As a result, both $t_{\text{total}}$ and $t_{\text{3D}}$ increases with decreasing $\tilde{n}$, leading them to become significantly longer than $t_{\text{min}}$.

If each round of 3D search is independent, the 3D search time should be $n_{\text{hits}}$ times the one round of 3D search time, displayed in the black curve.

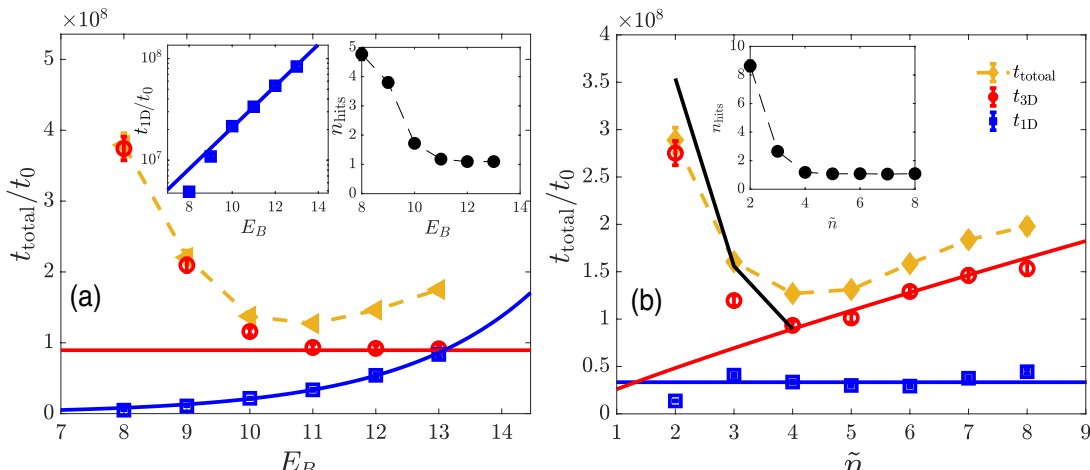

**Appendix 7—figure 2.** Search times as a function of $E_B$ and $\tilde{n}$. (a) $t_{total}$, $t_{3D}$, and $t_{1D}$ vs. at fixed $L^*$ and $\tilde{n}^*$. The red solid line is the prediction of $t_{3D}$ from **Equation 5**, and the blue line (also in the inset) is the exponential fit for $t_{1D}$. The black curve in the inset is the number of rounds the TF hits the antenna, $n_{hits}$, vs. $E_B$. (b) $t_{total}$, $t_{3D}$ and $t_{1D}$ vs. $\tilde{n}$ at fixed $L^*$ and $E_B^*$. The red and blue solid lines represent the predictions of $t_{3D}$ and $t_{1D}$ from **Equation 5**, respectively. $n_{hits}$ varying with $\tilde{n}$ is shown in the inset.

## Appendix 8

### A single round of 3D-1D search is typically the case in the optimal search process

In this section, we will show that the optimal search process (i.e. minimizing the mean first passage time) generically takes place via a single round of 3D-1D search. To do so, we first need to derive the expression for the mean total search time, $t_{\text{total}}$, and its dependence on the relevant parameters, including, importantly, $E_B$ and $L$. According to our model presented in the main text: A TF is initially in the 3D nucleus space; it can attach to the antenna and perform 1D diffusion with the coefficient denoted by $D_1$. We assume the DBD target is placed in the middle of the antenna, as shown in *Figure 1* of the main text. The mean search time taken for a TF to move from 3D space and attach to the antenna is denoted by $t_{3D}$. The TF can also detach from the antenna at a rate we denote by $\Gamma$, perform a 3D random walk, until ultimately attaching again to the antenna. In the following, we will make the assumption (often made in the context of FD model *Berg et al., 1981*; *Berg and von Hippel, 1987*; *Slutsky and Mirny, 2004*; *Mirny et al., 2009*; *Li et al., 2009*; *Bénichou et al., 2011*; *Sheinman et al., 2012*) that the binding position after such a 3D excursion is uniformly distributed on the antenna.

Both $D_1$ and $\Gamma$ are functions of $E_B$. When the TF is on the antenna, its mean search time to reach the DBD target, located a distance $x$ away on the antenna, is represented by $\mathcal{T}(x)$. Naturally, $\mathcal{T}(x)$ is zero when $x = 0$, the location of the DBD target. The mean total search time $t_{\text{total}}$ is thus given by

$$t_{\text{total}} = \frac{1}{L} \int_{-L/2}^{L/2} \mathcal{T}(x) dx + t_{3D}. \tag{22}$$

Consider a time interval $\Delta t$. With probability $\Gamma \Delta t$ the TF falls off, and the mean first passage time to reach the target is $t_{\text{total}} + \Delta t$. Otherwise, with probability $1 - \Gamma \Delta t$, the mean first passage time would be the average of the two possible outcomes, namely, $(\mathcal{T}(x + \Delta x) + \mathcal{T}(x - \Delta x))/2 + \Delta t$. This recursive reasoning leads us to the following equation:

$$\mathcal{T}(x) = \Delta t + \frac{\mathcal{T}(x - \Delta x) + \mathcal{T}(x + \Delta x)}{2} \left(1 - \Gamma \Delta t\right) + t_{\text{total}} \Gamma \Delta t. \tag{23}$$

In the continuous limit, we have

$$D_1 \frac{\partial^2 \mathcal{T}(x)}{\partial x^2} - \Gamma \mathcal{T}(x) + \Gamma t_{\text{total}} + 1 = 0, \tag{24}$$

where $D_1 \equiv (\Delta x)^2/(2\Delta t)$. By applying the boundary conditions, specifically the absorbing boundary condition at $x = 0$ and the reflecting boundary conditions at $x = \pm L/2$, $\mathcal{T}(x)$ is determined as

$$\mathcal{T}(x) = \left(t_{\text{total}} + \Gamma^{-1}\right) \left[1 - \frac{\cosh(\frac{L - 2|x|}{L_s})}{\cosh(\frac{L}{L_s})}\right], \tag{25}$$

where $L_s$, representing the 1D diffusion distance, is defined by $L_s \equiv 2\sqrt{D_1 \Gamma^{-1}}$ (*Wunderlich and Mirny, 2008*). Therefore, $t_{\text{total}}$ can be self-consistently solved by substituting $\mathcal{T}(x)$ into *Equation 22*, presented as follows:

$$t_{\text{total}} = \frac{L}{L_s} \coth\left(\frac{L}{L_s}\right) t_{3D} + \left[\frac{L}{L_s} \coth\left(\frac{L}{L_s}\right) - 1\right] \Gamma^{-1}. \tag{26}$$

We can obtain $t_{\text{total}}$ in two limiting cases:

- When $L \gg L_s$, where multiple rounds of 3D-1D excursions are needed to find the target, we have $t_{\text{total}} = \frac{L}{L_s} \left(t_{3D} + \Gamma^{-1}\right)$. This is the same as the results of the facilitated diffusion model (*Wunderlich and Mirny, 2008*; *Bénichou et al., 2011*; *Sheinman et al., 2012*; *Hachmo and Amir, 2023*).
- When $L \ll L_s$, where a single round of 3D-1D search can find the target, we get $t_{\text{total}} = t_{3D} + \frac{L^2}{12D_1}$. This matches *Equation 5* in the main text when $L < 2R$.

Note that the calculations above did not assume an optimal or efficient search process, but simply characterized the dependence of the mean first passage time on the model parameters. Next, based

on the expression for $t_{\text{total}}$ in **Equation 26**, we will prove that the search process involving multiple rounds of 3D-1D search is *not* optimal. This conclusion is reached by showing that the minimum $t_{\text{total}}$ always occurs in the regime where $L < 2L_s$ (since $L_s$ is the typical distance covered by 1D diffusion on the antenna before falling off, and the maximal distance needed to be traversed on the antenna is $L/2$). We emphasize an important distinction between our model and the well-studied facilitated diffusion model (where it is established that *multiple* rounds of search are needed for efficient search): in our case, the antenna length $L$ is a *variable* that can be optimized over (since we assume evolution has optimized over the size of the region flanking the DBD in which binding sites for the IDR are located). In the facilitated diffusion model, $L$ is assumed to be the length of the genome, since the interactions of the TF and the DNA are assumed non-specific. To proceed, let us first note that the factor $\coth\left(\dfrac{L}{L_s}\right)$ in **Equation 26** quickly converges to 1 as the ratio of $L/L_s$ increases. For instance, when the ratio is 2, we have $\coth(2) \approx 1.04$. Thus, for $L > 2L_s$, we could approximate $t_{\text{total}}$ as

$$t_{\text{total}} \approx \frac{L}{L_s} t_{3\text{D}} + \left(\frac{L}{L_s} - 1\right) \Gamma^{-1}. \tag{27}$$

Then, we consider how $t_{3\text{D}}$ varies with $L$. If the rate to hit the antenna after 3D diffusion was the sum of the rates to hit every infinitesimal segment of it, we would expect the rate to hit the antenna to be *linear* in $L$, and the mean first passage time to scale as $1/L$. However, these processes are not uncorrelated. Interestingly, when $L < 2R$, and assuming a straight antenna, $t_{3\text{D}}$ is proportional to $\ln(L)/L$ (see **Equation 5** in the main text), implying that for a straight antenna the effect of these correlations is mild. However, when $L > 2R$, the antenna is curved rather than straight (due to the limitations imposed on the DNA conformation by the dimensions of the nucleus), and the dependence of $t_{3\text{D}}$ on $L$ is weaker (**Hu et al., 2006**). Thus, for any given $L$ greater than $2L_s$, $t_{3\text{D}}$ varies more weakly than $1/L$, making the first term of $t_{\text{total}}$ in **Equation 27** an increasing function of $L$. Clearly, the second term increases as well. Therefore, when $L > 2L_s$, regardless of the functional forms of $D_1$ and $\Gamma^{-1}$ on $E_B$, $t_{\text{total}}$ decreases with decreasing $L$. Thus, the minimum $t_{\text{total}}$ occurs in the regime where $L < 2L_s$, and the optimal search typically consists of a single search round.

## Appendix 9

### Robustness of mean total search time with changing the antenna geometry

We checked that the mean total search time (Mean First Passage Time) for IDR targets, generated via the worm-like chain model – a random walk with a persistence length (*Milstein and Meiners, 2013*), does not significantly differ from the scenario where targets are arranged linearly, as presented in the main text. See *Appendix 9—figure 1* for details.

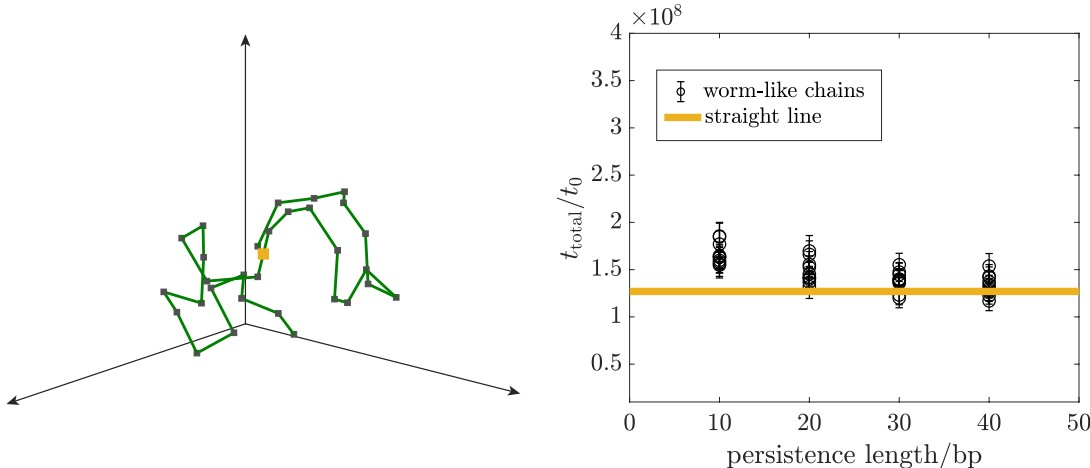

**Appendix 9—figure 1.** Left: an example of the target configuration (the antenna) generated by the worm-like chain model, where $L = 300$bp, $d = 10$bp, and the persistence length also equals 10bp. Right: Mean total search times weakly vary with persistence length when targets are generated using the worm-like chain model. The values are represented by the black dots, with parameters set to $E_B = 11$, $L = 300$ bp, and $\tilde{n} = 4$. For each persistence length, we obtained ten mean total search times by considering ten different target configurations. The mean total search times do not exhibit significant differences compared to that for targets arranged linearly (indicated by the horizontal orange line).

## Appendix 10

### MSD in a complex DNA configuration

In *Appendix 10—figure 1*, we run simulations to examine how the Mean Square Displacement (MSD) of a point searcher is influenced by the binding energy ($E_{\text{nonsp}}$) between the TF and the entire DNA in a complex DNA configuration. *Appendix 10—figure 1a* shows the DNA configuration generated via the worm-like chain model with a persistence length of 150 bp (*Milstein and Meiners, 2013*), where the volume fraction $\nu = 0.008$ is comparable to the experimental yeast DNA volume fraction. *Appendix 10—figure 1b* shows the MSD measured at different values of $E_{\text{nonsp}}$ for $\nu = 0.008$. When $E_{\text{nonsp}} \sim 1k_BT$, which corresponds to the non-specific binding energy regime in experiments (*Afek et al., 2014*), the effective 3D diffusion coefficient is slightly modified, which does not affect our conclusion on the optimal search process. Additionally, we observed that the MSD curve bends when $E_{\text{nonsp}}$ exceeds the typical range of non-specific binding energies.

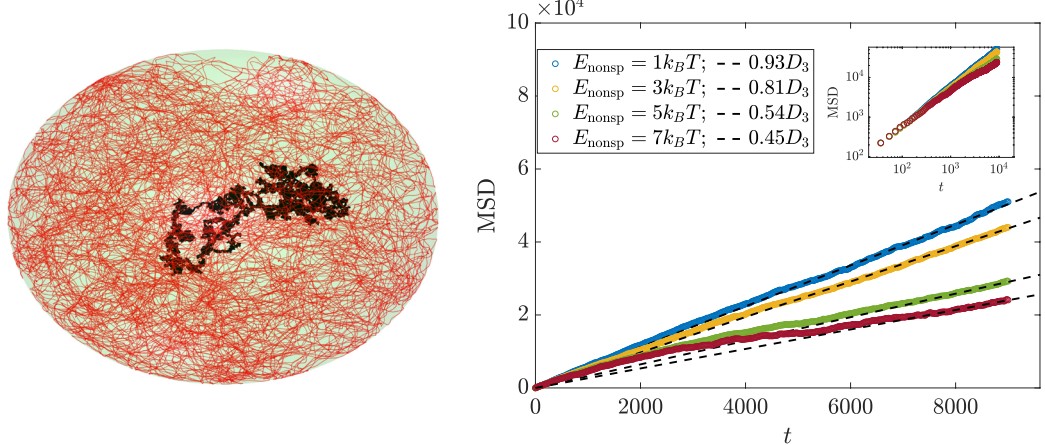

**Appendix 10—figure 1.** Search in a complex DNA configuration. (**a**) Configuration of DNA (in red) at $\nu = 0.008$ with a complex structure and a trajectory of TF (in black). A TF acts as a point searcher that diffuses in 3D space and becomes trapped once it hits the DNA, which is $1nm$ wide. It detaches at a rate of $\dfrac{D_3}{1nm^2} \exp(-E_{\text{nonsp}}/k_BT)$. (**b**) MSD at different $E_{\text{nonsp}}$.

## Appendix 11

### Properties of the octopusing walk

Once an IDR site binds to the antenna, the TF performs an octopusing walk, whereby different IDR binding sites constantly bind and unbind from the DNA. As we shall see, this will lead to an effective 1D diffusion of the TF. This indicates that during the octopusing walk, the TF neither gets trapped on a single target nor fully detaches from the antenna, allowing for rapid dynamics (in fact, the 1D diffusion coefficient can be comparable to the free 3D diffusion coefficient). We shall also show that the number of bound sites on the DNA is described by a probability distribution that is in steady state and obeys detailed balance. We denote that time-dependent probability distribution of the number of bound sites by $P(n, t)$ (with $n$ ranging from 0 to $\tilde{n}$). The dynamics of this probability distribution is given by:

$$\frac{dP(n,t)}{dt} = P(n-1,t)W_{n-1,n} + P(n+1,t)W_{n+1,n} + P(n,t)W_{n,n}, \tag{28}$$

where $W$ is called *transition rate matrix*, and $W_{n,n} = -W_{n,n-1} - W_{n,n+1}$. Intuitively, we anticipate that $W_{n,n-1}$, corresponding to the process of *detachment* of one of the sites, is *linear* in $n$, since we expect the different sites to be approximately independent of each other. Similarly, if we consider the different IDR sites as independent 'searchers', the on-rate (corresponding to $W_{n,n+1}$) to be linear in $(\tilde{n} - n)$. Indeed, Fig. *Appendix 11—figure 1a* corroborates this intuition to an excellent approximation. As the dynamics proceeds, *Equation 28* would reach a steady-state distribution governed by the matrix $W$. *Appendix 11—figure 1b* shows the excellent agreement between the steady-state associated with the matrix $W$ and that found directly from the MD simulation. This confirms that TF movement along the antenna is in a dynamic steady state. We also note that since our transition matrix is tri-diagonal, it satisfies the detailed balance condition, that is, in the steady-state reached, we have: $P(n)W_{n,n-1} = P(n-1)W_{n-1,n}$, where $P(n)$ is the steady-state distribution. A priori, one may think that the TF is facing two contradictory demands: on the one hand, it has to perform rapid diffusion along the DNA. On the other hand, it needs to bind sufficiently strongly so that it does not fall off the DNA too soon. Indeed, this is known as the speed-stability paradox within the context of TF search in bacteria. Importantly, the simple picture we described above explains an elegant mechanism to bypass this paradox, achieving a high diffusion coefficient with a low rate of detaching from the DNA. This comes about due to the compensatory nature of the dynamics outlined above: there is, in effect, a feedback mechanism whereby if only a few sites are attached to the DNA, the on-rate increases, and, similarly, if too many are attached (thus prohibiting the 1D diffusion), the off-rate increases. This is manifested by the peaked steady-state distribution shown in *Appendix 11—figure 1b*. In fact, this mechanism is remarkably effective even when the typical (i.e., most probable) number of binding sites is as small as 2 (note that in this case, as shown in the inset of *Appendix 11—figure 1a*, the TF may cover a distance of more than 100 bp before falling off).

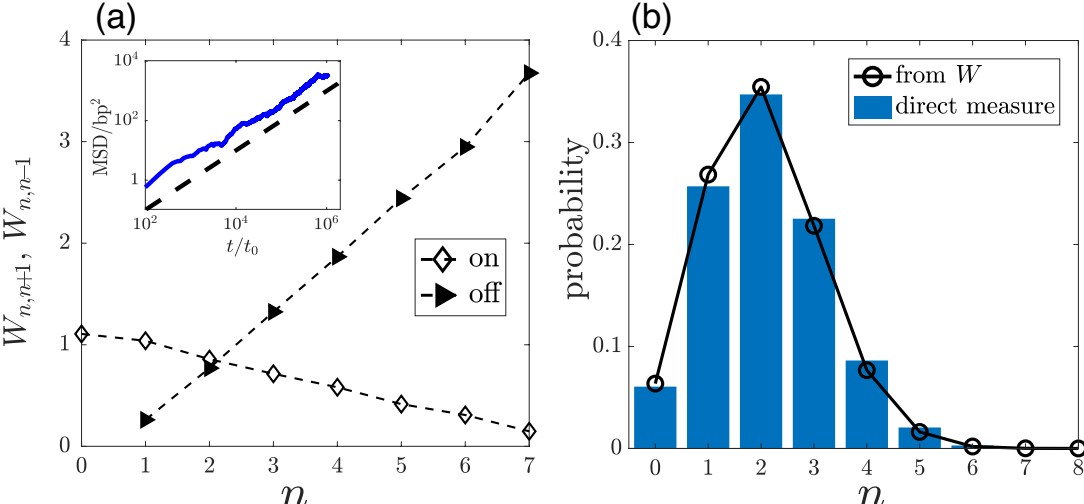

**Appendix 11—figure 1.** Property of 1D octopusing walk. (**a**) $W_{n,n+1}$ (or $W_{n,n-1}$) denoted by 'on' (or 'off') varies with $n$ for $E_B = 6$ and $\tilde{n} = 12$. Inset: Mean square displacement (MSD) suggests diffusive motion, as indicated by the dashed line following $\text{MSD} \propto t$. (**b**) Steady probability distribution of bound sites.

