## [Editor Report · eLife Assessment]

This paper presents an **important** theoretical exploration of how a flexible protein domain with multiple DNA binding sites may simultaneously provide stability to the DNA-bound state and enables exploration of the DNA strand. The authors propose a mechanism ("octopusing") for protein doing a random walk while bound to DNA which simultaneously enables exploration of the DNA strand and enhances the stability of the bound state. This study presents **compelling** evidence that their findings has implications for the way intrinsically disordered regions (IDR) of transcription factors proteins (TF) can enhance their ability to efficiently find their binding site on the DNA from which they exert control over the transcription of their target gene. The paper concludes with a comparison of model predictions with experimental data which gives further support to the proposed model.

---

## [Referee Report · Reviewer #1 (Public review)]

Summary:

The authors define the principles that, based on first principles, should be guiding the optimisation of transcription factors with intrinsically disordered regions (IDR). The authors introduce an original search process, coined "octopusing", that involves transcription factor IDR and their binding affinities to optimise search times and binding affinities. The first part concerns the optimal strategies to define binding affinities to the genome in the receiving region that is called the "antenna", highlighting the following: (i) reduce the target to IDR-binding distance on the genome, (ii) optimise the distance between the DNA binding domain and the binding sites on the IDR to be as close as possible to the distance between their binding sites on the genome; (iii) keep the same number of binding sites and their targets and modulate this number with binding strength, reducing them with increased strength; (iv) modulate the binding strength to be above a threshold that depends on the proportion of IDR binding sites in the antenna. The second part concerns the scaling of the search time in function of key parameters such as the volume of the nucleus, and the size of the antenna, derived as a combination of 3D search and 1D "octopusing". The third part focuses on validation, where the current results are compared to binding probability data from a single experiment, and new experiments are proposed to further validate the model as well as testing designed transcription factors.

Strengths:

The strength of this work is that it provides simple, interpretable and testable theoretical conclusions. This will allow the derived design principles to be understood, evaluated and improved in the future. The theoretical derivations are rigorous. The authors provide a comparison to experiments, and also propose new experiments to be performed in the future. This is a great value in the paper since it will set the stage and inspire new experimental techniques. Further, the field needs inspiration and motivation to develop these techniques, since they are required to benchmark the transcription factors designed with the methods presented in this paper, as well as to develop novel data based or in vivo methods that would greatly benefit the field. As such, this paper is a fundamental contribution to the field.

Weaknesses:

The model presents many first principles to drive the design of transcription factors, but arguably, other principles and mechanisms might also play a role by being beneficial to the search and binding process. These other principles are mentioned at the end of the discussion part of the paper. On the other hand, an important task left to do, is to critically consider these principles altogether, and analyse the available data to quantify which role is predominant among transcription factors IDRs functions. Further, since one function doesn't exclude another, a theoretical investigation of possible crosstalk, interaction, and cooperativity of those different hypothetical functions is still missing.

---

## [Referee Report · Reviewer #2 (Public review)]

Summary:

This is an interesting theoretical exploration of how a flexible protein domain, which has multiple DNA-binding sites along it, affects the stability of the protein-DNA complex. It proposes a mechanism ("octopusing") for protein doing a random walk while bound to DNA which simultaneously enables exploration of the DNA strand and stability of the bound state.

Strengths:

Stability of the protein-DNA bound state and the ability of the protein to perform 1d diffusion along the DNA are two properties of a transcription factor that are usually seen as being in opposition of each other. The octopusing mechanism is an elegant resolution of the puzzle of how both could be accommodated. This mechanism has interesting biological implications for the functional role of intrinsically disordered domains in transcription factor (TF) proteins. They show theoretically how these domains, if flexible and able to make multiple weak contacts with the DNA, can enhance the ability of the TF to efficiently find their binding site on the DNA from which they exert control over the transcription of their target gene. The paper concludes with a comparison of model predictions with experimental data which gives further support to the proposed mechanism. Overall, this is an interesting and well-executed theoretical paper that proposes an interesting idea about the functional role for IDR domains in TFs.

Weaknesses:

It is not clear how ubiquitous among eukaryotic transcription factors are the DNA binding sites for multiple subdomains along the IDR, which are assumed by the model. These assumptions though, provide interesting points of departure for further experiments.

---

## [Author Response]

The following is the authors’ response to the original reviews

**Reviewer #1 (Public review):**
Summary:The authors define the principles that, based on first principles, should be guiding the optimisation of trascription factors with intrinsically disordered regions (IDR). The first part of the study defines the following principles to optimize the binding affinities to the genome in the receiving region that is called the ”antenna”: (i) reduce the target to IDR-binding distance on the genome, (ii) optimise the distance betwee the DNA binding domain and the binding sites on the IDR to be as close as possible to the distance between their binding sites on the genome; (iii) keep the same number of binding sites and their targets and modulate this number with binding strength, reducing them with increased strenght; (iv) modulate the binding strenght to be above a threshold that depends on the proportion of IDR binding sites in the antenna. The second part defines the scaling of the seach time in function of key parameters such as the volume of the nucleus, and the size of the antenna, derived as a combination of 3D search of the antenna and 1D ”octopusing” on the antenna. The third part focuses on validation, where the current results are compared to binding probabilith data from a single experiment, and new experiment are proposed to further validate the model as well as testing designed transcription factors.Strengths:The strength of this work is that it provides simple, interpretable and testable theoretical conclusions. This will allow the derived design principles to be understood, evaluated and improved in the future. The theoretical derivations are rigorous. The authors provides a comparison to experiments, and also propose new experiments to be performed in the future, this is a great value in the paper since it will set the stage and inspire new experimental techniques. Further, the field needs inspiration and motivations to develop these techniques, since they are required to benchmark the transcription factors designed with the methods presented in this paper, as well as to develop novel data based or in vivo methods that would greatly benefit the field. As such, this paper is a fundamental contribution to the field.Weaknesses:The model assumption that the interaction between the transcription factor and the DNA outside of the antenna region is negligible is probably too strong for many/most transcription factors, particularly in organisms with a longer genome than yeasts. The model presents many first principles to drive the design of transcription factor, but arguably, other principles and mechanisms might also play a role by being beneficial to the search and binding process. Specifically: (i) a role of the IDR in complex formation and cooperativity between multiple trascription factors, (ii) ability of the IDR to do parallel searching based on multiple DNA binding sites spaced by disordered regions, (iii) affinity of the IDR to specific compartmentalisations in the nucleus reducing the search time, etc. The paper would be improved by a discussion over alternative mechanisms.

We thank the reviewer for highlighting that our work delivers simple, interpretable and rigorously derived conclusions, backed by experimental comparison and concrete proposals for future studies.

Regarding interactions outside the antenna region, Supplementary S10 shows that the non-specific IDR–DNA interactions (on the order of 1 kBT) only slightly alter the 3D diffusion coefficient and thus do not affect our conclusions regarding the optimal search process.

We have also added sentences in the discussion section regarding the alternative mechanism.

**Reviewer #2 (Public review):**
Summary:This is an interesting theoretical exploration of how a flexible protein domain, which has multiple DNAbinding sites along it, affects the stability of the protein-DNA complex. It proposes a mechanism (”octopusing”) for protein doing a random walk while bound to DNA which simultaneously enables exploration of the DNA strand and stability of the bound state.Strengths:Stability of the protein-DNA bound state and the ability of the protein to perform 1d diffusion along the DNA are two properties of a transcription factor that are usually seen as being in opposition of each other. The octopusing mechanism is an elegant resolution of the puzzle of how both could be accommodated. This mechanism has interesting biological implications for the functional role of intrinsically disordered domains in transcription factor (TF) proteins. They show theoretically how these domains, if flexible and able to make multiple weak contacts with the DNA, can enhance the ability of the TF to efficiently find their binding site on the DNA from which they exert control over the transcription of their target gene. The paper concludes with a comparison of model predictions with experimental data which gives further support to the proposed model. Overall, this is an interesting and well executed theoretical paper that proposes an interesting idea about the functional role for IDR domains in TFs.Weaknesses:IDR domains are assumed flexible which I believe is not always the case. Also, I’m not sure how ubiquitous are the assumed binding sites on the DNA for multiple subdomains along the IDR. These assumptions though seem like interesting points of departure for further experiments.

We thank the reviewer for their careful and insightful evaluation of our work. In particular, we appreciate your emphasis on the inherent trade-off between binding stability and one-dimensional diffusion, and your recognition of how the octopusing mechanism elegantly reconciles these conflicting requirements.

To address the flexibility of TFs with IDRs, we incorporated the spring’s rest length—effectively introducing tunable rigidity—in Supplementary Section S1, and we show that our design principles for binding probability remain robust. Indeed, this is a highly interesting point; a comprehensive study will require more detailed modeling alongside experimental validation.

We acknowledge that the current evidence for IDR-directed DNA binding is primarily derived from a limited number of well-studied cases, particularly Msn2 in yeast, and the ubiquity of this mechanism across diverse transcription factors remains to be established.

**Reviewer #1 (Recommendations for the authors):**
The paper jumps to fast to the results, an larger introduction might improve the paper, the current introduction jumps too fast to results. Further, line 50, I don’t think that the figure is properly referenced. The formula 2 is confusing since what is the target volume V1 is not explained in the context of the formula, please expand the explanations.

We appreciate the reviewer’s valuable recommendations. We have expanded the Introduction, clarified *V*_1_, and updated the line 50.

**Reviewer #2 (Recommendations for the authors):**
I have some mostly minor suggestions to the authors for improving the manuscript:In the abstract and introduction on at least two occasions the authors talk about IDRs as though they’re necessarily flexible. My understanding is that, while this is a very reasonable assumption, I don’t think this is something we know with any certainty for most IDRs. If the authors agree with my assessment I think they should reflect this uncertainty in the writing.

Thank you for the recommendations. We revised the wording to reflect the uncertainty, changing it to: “... commonly assumed to behave as a long, flexible...” and “...can be assumed as flexible....”.

It took me a bit of time to figure out what’s going on in Figure 1b. To help the reader I would suggest labeling the DBD targets (yellow square) and the IDR targets (gray squares) as such. The figure also left me guessing whether the DBD domain can bind to the IDR targets non-specifically? (I presume not.) This also brought a slightly bigger question into focus for me, wouldn’t the presence of the IDR binding ”sites” (since these ”sites” are on the protein I think the term ”domains” instead of ”sites”) mean that this would increase the time the protein is bound non-specifically somewhere far from the target thereby increasing the search time. Or is the ability of the protein to bind specifically to DNA away from the DBD target ignored?

We have labeled the DBD targets and IDR targets in the figure. ‘Domains’ usually refers to structured parts; we keep using ‘sites’ and clarify that they correspond to short linear motifs.

The reviewer is correct. Our model omits any non-specific binding between the DBD and IDR-binding targets, as well as between the TF and other DNA regions. If such interactions were to substantially lengthen the search time, they would effectively revert our mechanism to the classical bacterial facilitateddiffusion model, which is generally considered inappropriate for IDR-mediated TF search in eukaryotic cells. However, Supplementary Figure S10 demonstrates that non-specific IDR–DNA interactions induce only marginal changes in the effective three-dimensional diffusion coefficient within complex chromatin environments, and therefore do not alter our conclusions regarding the optimal search process.

In Equation 2 and the text that follows I was left wondering what is the target volume V1. Also, I think it would be helpful to the reader to give them a sense of scale for the dimension full quantities appearing in Equation 2. This is done later when comparing the theory to experimental data, but I think it would be helpful to give a sense of size earlier in the manuscript.

*V*_1_ denotes the volume of the IDR–binding target region, which is on the order of bp^3^. *f*(*d,l*_0_) has units of inverse volume. We have included the units and specified the order of magnitude of *V*_1_ after Equation 2.

The binding energy EB is discussed a number of times but it wasn’t clear to me that this quantity referred to the energy per IDR site on the DNA or the total energy when the IDR is bound to DNA. In Figure 1 it would seem that the model allows only one IDR domain bound at a given time but I think the model allows for multiple IDR domains to be bound to the IDR target sites simultaneously. Right? Maybe make this clear in the Figure and the text.

*E*_B_ denotes the binding energy per binding site, where each site corresponds to a short linear motif. Yes, we allow for multiple IDR domains to be bound to the IDR target sites simultaneously. We have clarified the definition of *E*_B_ and adjusted the figure slightly to avoid any misunderstanding.

After Eq 4 the discussion suggests that for ϕ << 1 the threshold energy is much greater than kBT, but that’s hard to imagine given that the logarithmic dependence of the latter on the former. Also in Figure 2d it seems that the threshold energy is about 8 kBT. Clearly this is not a big deal, just thought the authors might want to revise the language.

Thank you. We now clarify the sentence using the representative values of ϕ and *E*_th_ after Equation 4.

Right after Figure 2 there is a discussion of the different parameters that the authors vary. I suggest having a figure that illustrates these parameters (possibly in Figure 1b) to make it easier to follow the discussion.

We have added explanations of the relevant parameters in Figure 1 for clarity.

When discussing the dynamics of search the result stated is that the search time is minimum for a specific value of R. I think it would be useful to translate this into a TF concentration. Also, if R represents the radius of the cells nucleus 1/6 um is almost an order of magnitude smaller than the size of a typical nucleus. Is this a worry? Either way some clarification of this number would be helpful.

Thank you for the suggestion. As noted later in this section, we have translated R into an equivalent TF concentration, and we clarify that we assume the scaling of the minimum search time remains unchanged when extrapolated to the size of a typical nucleus.

There is a comment regarding the role of the DNA persistence length and how it was not accounted for. It would be helpful if the authors could add a sentence or two explains how a folded DNA conformation, as is the case in the nucleus, would affect their calculation. (So that the reader gets an idea without having to get into the details described in the Supplement).

Thank you. We have revised the sentence to: “We have verified that reducing the DNA persistence length, which promotes increased DNA coiling, results in only a modest increase in mean search time. Even under extreme coiling conditions, the increase remains below 30% of the baseline value, as detailed in Supplementary S9.”.